# The GGCMI Phase 2 emulators: global gridded crop model responses to changes in CO$_2$, temperature, water, and nitrogen (version 1.0)

James A. Franke[1,2], Christoph Müller[3], Joshua Elliott[2,4], Alex C. Ruane[5], Jonas Jägermeyr[4,2,3,5], Abigail Snyder[6], Marie Dury[7], Pete D. Falloon[8], Christian Folberth[9], Louis François[7], Tobias Hank[10], R. Cesar Izaurralde[11,12], Ingrid Jacquemin[7], Curtis Jones[11], Michelle Li[2,13], Wenfeng Liu[14,15], Stefan Olin[16], Meridel Phillips[5,17], Thomas A. M. Pugh[18,19], Ashwan Reddy[11], Karina Williams[8,20], Ziwei Wang[1,2], Florian Zabel[10], and Elisabeth J. Moyer[1,2]

[1]Department of the Geophysical Sciences, University of Chicago, Chicago, IL, USA
[2]Center for Robust Decision-making on Climate and Energy Policy (RDCEP), University of Chicago, Chicago, IL, USA
[3]Potsdam Institute for Climate Impact Research, Member of the Leibniz Association, Potsdam, Germany
[4]Department of Computer Science, University of Chicago, Chicago, IL, USA
[5]NASA Goddard Institute for Space Studies, New York, NY, United States
[6]Joint Global Change Research Institute, Pacific Northwest National Laboratory, College Park, MD, USA
[7]Unité de Modélisation du Climat et des Cycles Biogéochimiques, UR SPHERES, Institut d'Astrophysique et de Géophysique, University of Liège, Belgium
[8]Met Office Hadley Centre, Exeter, United Kingdom
[9]Ecosystem Services and Management Program, International Institute for Applied Systems Analysis, Laxenburg, Austria
[10]Department of Geography, Ludwig-Maximilians-Universität, Munich, Germany
[11]Department of Geographical Sciences, University of Maryland, College Park, MD, USA
[12]Texas Agrilife Research and Extension, Texas A&M University, Temple, TX, USA
[13]Department of Statistics, University of Chicago, Chicago, IL, USA
[14]EAWAG, Swiss Federal Institute of Aquatic Science and Technology, Dübendorf, Switzerland
[15]Laboratoire des Sciences du Climat et de l'Environnement, LSCE/IPSL, CEA-CNRS-UVSQ, Université Paris-Saclay, F-91191 Gif-sur-Yvette, France.
[16]Department of Physical Geography and Ecosystem Science, Lund University, Lund, Sweden
[17]Earth Institute Center for Climate Systems Research, Columbia University, New York, NY, USA
[18]School of Geography, Earth and Environmental Sciences, University of Birmingham, Birmingham, UK.
[19]Birmingham Institute of Forest Research, University of Birmingham, Birmingham, UK.
[20]Global Systems Institute, University of Exeter, Laver Building, North Park Road, Exeter, EX4 4QE, UK

**Correspondence:** James Franke (jfranke@uchicago.edu)

**Abstract.** Statistical emulation allows combining advantageous features of statistical and process-based crop models for understanding the effects of future climate changes on crop yields. We describe here the development of emulators for nine process-based crop models and five crops using output from the Global Gridded Model Intercomparison Project (GGCMI) Phase 2. The GGCMI Phase 2 experiment is designed with the explicit goal of producing a structured training dataset for emulator development that samples across four dimensions relevant to crop yields: atmospheric carbon dioxide (CO$_2$) concentrations, temperature, water supply, and nitrogen inputs (CTWN). Simulations are run under two different adaptation assumptions: that growing seasons shorten in warmer climates, and that cultivar choice allows growing seasons to remain fixed. The dataset

allows emulating the climatological mean yield response of all models with a simple polynomial in mean growing-season values. Climatological mean yields are a central metric in climate change impact analysis; we show here that they can be captured

without relying on interannual variations. In general, emulation errors are negligible relative to differences across crop models or even across climate model scenarios; errors become significant only in some marginal lands where crops are not currently grown. We demonstrate that the resulting GGCMI emulators can reproduce yields under realistic future climate simulations, even though the GGCMI Phase 2 dataset is constructed with uniform CTWN offsets, suggesting that the effects of changes in temperature and precipitation distributions are small relative to those of changing means. The resulting emulators therefore

capture relevant crop model responses in a lightweight, computationally tractable form, providing a tool that can facilitate model comparison, diagnosis of interacting factors affecting yields, and integrated assessment of climate impacts.

## 1   Introduction

Improving our understanding of the impacts of future climate change on crop yields is critical for global food security in the twenty-first century. Projections of future yields under climate change are generally made with one of two approaches: either

process-based models, which simulate the process of photosynthesis and the biology and phenology of individual crops, or statistical models, which use historical weather and yield data to capture relationships between observed crop yields and major drivers. Process-based crop models provide some advantages, including capturing the direct effects of $CO_2$ fertilization and allowing projections in areas where crops are not currently grown. However, they are computationally expensive, and can be difficult or impossible to directly integrate into integrated climate change impacts assessments. Statistical crop models can only

capture crop responses under the range of current conditions but have several advantages: they implicitly include management and behavioral practices that are difficult to model explicitly, and they are typically simple analytical expressions that are easily implemented by downstream impact modelers. Both types of models are routinely used, and comparative studies have concluded that when done carefully, both approaches can provide similar yield estimates (e.g. Lobell and Burke, 2010; Moore et al., 2017; Roberts et al., 2017; Zhao et al., 2017; Liu et al., 2016a).

Statistical emulation allows combining some of the advantageous features of both statistical and process-based models. The approach involves constructing a "surrogate model" of numerical simulations by using their output as training data for a statistical representation (e.g. O'Hagan, 2006; Conti et al., 2009). Emulation is particularly useful in cases where simulations are complex and output data volumes are large and has been used in a variety of fields, including hydrology (e.g. Razavi et al., 2012), engineering (e.g. Storlie et al., 2009), environmental sciences (e.g. Ratto et al., 2012), and climate (e.g. Castruccio

et al., 2014; Holden et al., 2014). For agricultural impacts studies, emulation of process-based models allows capturing key relationships between input variables in a lightweight, flexible form that is compatible with economic studies. The resultant statistical model can produce yield projections under arbitrary emissions scenarios and is an important diagnostic tool for model comparison and model evaluation.

Interest is rising in applying statistical emulation to crop models, and multiple studies have developed crop model emulators

in the past decade. Early studies proposing or describing potential crop yield emulators include Howden and Crimp (2005);

Räisänen and Ruokolainen (2006); Lobell and Burke (2010), and Ferrise et al. (2011). Studies developing single-model emulators include Holzkämper et al. (2012) for the CropSyst model, Ruane et al. (2013) for the CERES wheat model, and Oyebamiji et al. (2015) for the LPJmL model. More recently, emulators have begun to be used in the context of multi-model intercomparison, with multiple authors (Blanc and Sultan, 2015; Blanc, 2017; Ostberg et al., 2018; Mistry et al., 2017) using them to analyze the five crop models of the Inter-Sectoral Impact Model Intercomparison Project (ISIMIP). ISIMIP offers a relatively large training set – control, historical, and several Representative Concentration Pathway (RCP) scenarios using output from up to five climate models (Warszawski et al., 2014; Frieler et al., 2017) – and choices of emulation strategy differ. Blanc and Sultan (2015) and Blanc (2017) use historical and RPC8.5 scenarios, combine multiple climate model projections for RCP8.5, and regress across soil regions. Ostberg et al. (2018) use global mean temperature change (and $CO_2$) as regressors, and then pattern-scales to emulate local yields. Mistry et al. (2017) compare emulated and observed historical yields, using local weather data and a historical crop simulation. The constraints of the ISIMIP experiment mean that all these efforts do share important common features. All emulate annual crop yields along an entire scenario or scenarios, and all future climate scenarios are non-stationary, with important covariates (temperature and precipitation for example) evolving simultaneously.

An alternative approach to emulation involves construction of a "parameter sweep" training set, a collection of multiple stationary scenarios that systematically cover a range of input parameter values. A parameter sweep offers several important advantages for emulation over an experiment in which climate evolves over time. First, it allows separating the effects of different variables that affect yields but that are highly correlated in realistic future scenarios like those used in ISIMIP (e.g. $CO_2$ and temperature). Second, it allows making a distinction between year-to-year yield variations and climatological changes, which may involve different responses to the particular climate regressors used (e.g. Ruane et al., 2016). For example, if year-to-year yield variations are driven predominantly by variations in the distribution of temperatures throughout the growing period, and long-term climate changes are driven predominantly by additive mean shifts, then regressing on the mean growing period temperature will produce different yield responses at annual vs. climatological timescales.

Systematic parameter sweeps have begun to be used in crop model evaluation and emulation, with early efforts in 2014 and 2015 (Ruane et al., 2014; Makowski et al., 2015; Pirttioja et al., 2015), and several recent studies in 2018 and 2019 (Fronzek et al., 2018; Ruiz-Ramos et al., 2018; Snyder et al., 2019). These three studies sample multiple perturbations to temperature and precipitation, and two of the three add $CO_2$ as well, for a total of 132, 99, and 220 different combinations, respectively. All take advantage of the structured training set to construct emulators ("response surfaces") of climatological mean yields, omitting year-to-year variations. All the 2018–2019 papers have some limitations, however, for assessing global agricultural impacts, including that none evaluate responses in every grid cell globally. Two involve many crop models but only one crop (wheat) (Fronzek et al., 2018; Ruiz-Ramos et al., 2018) and cover only 1-4 individual sites. Snyder et al. (2019) analyzes five crops over ~1,000 sites with individual site-specific crop models, and extrapolates in space to estimate mean latitudinal responses.

In this paper we describe a set of globally-gridded crop model emulators developed from the new parameter-sweep dataset of the Global Gridded Crop Model Intercomparison (GGCMI) Phase 2 effort. GGCMI Phase 2, a part of the Agricultural Model Intercomparison and Improvement Project (AgMIP) (Rosenzweig et al., 2013, 2014), provides the first near-global-coverage systematic parameter sweep of multi-model crop simulations consisting of up to 756 combinations in $CO_2$, temperature, water

supply, applied nitrogen, and two different assumptions on growing season adaptation ("A0": none and "A1": retaining growing season length) (CTWN-A, Franke et al., 2020; Minoli et al., 2019b). The experiment is designed to allow diagnosing the impacts on crop yields of both individual factors and their joint effects, and to allow construction of crop model emulators. In Section 2 following, we describe the training dataset, including the GGCMI Phase 2 experimental protocol and model participation (Section 2.1) and the models' differing year-to-year and climatological mean responses (Section 2.2). Section 3 describes the statistical model used for emulation, Section 4 evaluates measures of emulator fidelity, and Section 5 shows examples of preliminary results.

## 2 Training dataset

### 2.1 The GGCMI Phase 2 dataset

The GGCMI Phase 2 simulations are described in detail in Franke et al. (2020), but we summarize briefly here. The experiment involves 9 different globally gridded crop models, each simulating multiple crops (maize, rice, soybean, and spring and winter wheat) across a systematic parameter sweep of as many as 756 combinations, each driven by a historical climate timeseries with systematic perturbations to $CO_2$, temperature, water supply, and nitrogen application (CTWN). The simulation protocol involves 4 levels of atmospheric $CO_2$, 7 of temperature, 9 of water supply, and 3 of applied nitrogen, and simulations are repeated for 2 adaptation scenarios: "A0" simulations assume no adaptation in cultivar choice, so that growing seasons shorten in warmer climates, and "A1" simulations assume that adaptation in cultivar choice maintains fixed growing seasons. The complete protocol for each modeling group involves up to 43,524 years of global simulated output for each crop. Because the computational demand is high, modeling groups were allowed to submit at various specified levels of participation, with the lowest recommended level of participation consisting of 20% of the maximum possible simulations. The mean participation level is 65%, but three models (APSIM-UGOE, EPIC-IIASA, and ORCHIDEE-crop) contributed data below the recommended threshold ($< 5\%$ of the full protocol) and are excluded here since they could not be robustly emulated. Table 1 shows the participating models and the number of simulation scenarios that each provides, and Supplemental Figure S1 shows model sampling density. See Franke et al. (2020) for the parameter combinations included by each model. Table 2 shows the specified input values; we sample across all parameter combinations.

Each individual crop model simulation is run for 31 years over historic weather for the period of 1981-2010, with added uniform perturbations to any of the CTWN variables. Historical weather is taken for most models from the AgMERRA (Ruane et al., 2015) historical daily climate data product, but the PROMET model uses the ERA-Interim reanalysis (Dee et al., 2011) and the JULES model uses a bias-corrected version of ERA-Interim, WFDEI (WATCH-Forcing-Data-ERA-Interim, Weedon et al., 2014) as these groups have specific sub-daily input data requirements. Temperature perturbations are applied as additive mean shifts, water supply as fractional multipliers to precipitation (except in the irrigated $W_\infty$ case), and $CO_2$ and nitrogen application levels are specified as fixed values. Models provide near-global output at 0.5 degree latitude and longitude resolution for each simulation year, including areas not currently cultivated. Crop models included here are not formally calibrated, given that there is no adequate calibration target for gridded global-scale crop model simulations. This may be a shortcoming if

**Table 1.** Crop models included in GGCMI Phase 2 emulators and the number of CTWN-A (Carbon dioxide, Temperature, Water, Nitrogen, Adaptation) simulations performed for each model. The maximum number is 756 for A0 (no adaptation) experiments, and 648 for A1 (maintaining growing season length) experiments since T0 is not simulated under A1. "N-Dim." indicates whether the models are able to represent varying nitrogen levels. Each model provides the same set of CTWN simulations across all its modeled crops, but some models omit individual crops. Table adapted from Franke et al. (2020). For clarity, three simulation models that submitted data to the GGCMI Phase 2 experiment (Franke et al., 2020) are not shown here, as they provided a training set too small to be used in emulation.

| Model (Key Citations) | Maize | Soybean | Rice | Winter wheat | Spring wheat | N dim. | Sims per crop (A0 / A1) |
|---|---|---|---|---|---|---|---|
| **CARAIB**, Dury et al. (2011); Pirttioja et al. (2015) | X | X | X | X | X | – | **252 / 216** |
| **EPIC-TAMU**, Izaurralde et al. (2006) | X | X | X | X | X | X | **756 / 648** |
| **JULES**, Osborne et al. (2015); Williams and Falloon (2015); Williams et al. (2017) | X | X | X | – | X | – | **252 / 0** |
| **GEPIC**, Liu et al. (2007); Folberth et al. (2012) | X | X | X | X | X | X | 430 / 181 |
| **LPJ-GUESS**, Lindeskog et al. (2013); Olin et al. (2015) | X | – | – | X | X | X | **756 / 648** |
| **LPJmL**, von Bloh et al. (2018) | X | X | X | X | X | X | **756 / 648** |
| **pDSSAT**, Elliott et al. (2014); Jones et al. (2003) | X | X | X | X | X | X | **756 / 648** |
| **PEPIC**, Liu et al. (2016b, c) | X | X | X | X | X | X | 149 / 121 |
| **PROMET**, Hank et al. (2015); Mauser et al. (2015); Zabel et al. (2019) | X | X | X | X | X | – | 261 / 232 |

targeting absolute yield levels, but when focusing on relative yield changes, calibration can also have negative effects on model
skill (Müller et al., 2017). In analyses where we distinguish yields over currently cultivated land, we use the harvested area
masks of Portmann et al. (2010). (See Supplemental Figure S2 for maps of cultivated area.)

### 2.2 Climatological vs. year-to-year responses

The central metric in assessments of climate change impacts on crop yields is the change in multi-annual means (e.g. Schlenker
and Roberts, 2009; Challinor et al., 2014; Rosenzweig et al., 2014; Müller et al., 2015; Zhao et al., 2016; Hsiang et al.,
2017). Agricultural impacts assessments work with multi-annual yields, as their analysis frameworks require information
on long-term effects (e.g. Nelson et al., 2014b; Stevanović et al., 2016; Wiebe et al., 2015; Hasegawa et al., 2018; Snyder

**Table 2.** GGCMI Phase 2 input levels for the parameter sweep. Values for temperature and water supply are perturbations from the historical climatology. For water supply, perturbations are fractional changes to historical precipitation, except in the irrigated ($W_\infty$) simulations, which are all performed with the maximum beneficial levels of water. Bold font indicates the 'baseline' historical level. The full protocol samples across all parameter combinations for a total of 756 cases. Table repeated from Franke et al. (2020).

| Input variable | Tested range | Unit |
|---|---|---|
| $[CO_2]$ (C) | **360**, 510, 660, 810 | ppm |
| Temperature (T) | -1, **0**, 1, 2, 3, 4, 6 | °C |
| Precipitation (W) | -50, -30, -20, -10, **0**, 10, 20, 30, (and $W_\infty$) | % |
| Applied nitrogen (N) | 10, 60, **200** | kg ha$^{-1}$ |
| Adaptation (A) | **A0: none**, A1: new cultivar to maintain original growing season length | - |

et al., 2019). Changes in extremes or year-to-year variability are other metrics of potential interest, but are often not explicitly considered in integrated climate change impact assessments or land-use change projections. For this reason we emulate the climatological mean response, i.e. the change in aggregated mean yield in each 30-year simulation. Emulation then becomes
relatively straightforward since changes in time-averaged yields are considerably smoother than those in year-to-year yield response. In the GGCMI Phase 2 simulation output dataset, year-to-year responses to weather are also often quantitatively distinct from responses to climatological shifts, with the discrepancy especially strong in wheat and rice. The difference in behavior is illustrated in Figure 1, which shows irrigated and rainfed maize and wheat in representative locations. When discrepancies are large, year-to-year responses are generally stronger than climatological ones, but exact responses differ by
crop and region and even by model within GGCMI Phase 2.

While differences in responses at different timescales can arise for many reasons, including memory in the crop model or lurking covariates, the most likely explanation here is that the regressors used, mean growing-season temperature or precipitation, do not fully describe the conditions that affect crop yields. The mean growing-season value is only a proxy for the distribution of daily climatic conditions that crops are sensitive to, and present-day variations between years can be very dif-
ferent from future forced changes. Present-day variations in growing season *means* from year to year may be associated with changes in growing season *distributions* that are unrelated to changes in future warmer climates: that is, a warm year at present may be quite different from a warm year in the future (e.g. Ruane et al., 2016). Changes in temperature distributions have been shown to strongly affect crop yields (e.g. Hansen and Jones, 2000; Gadgil et al., 2002) though precipitation effects should be smaller since crops respond not to rainfall but to soil moisture, which integrates over weeks or even months (e.g. Potter et al.,
2005; Glotter et al., 2014; Challinor et al., 2004).

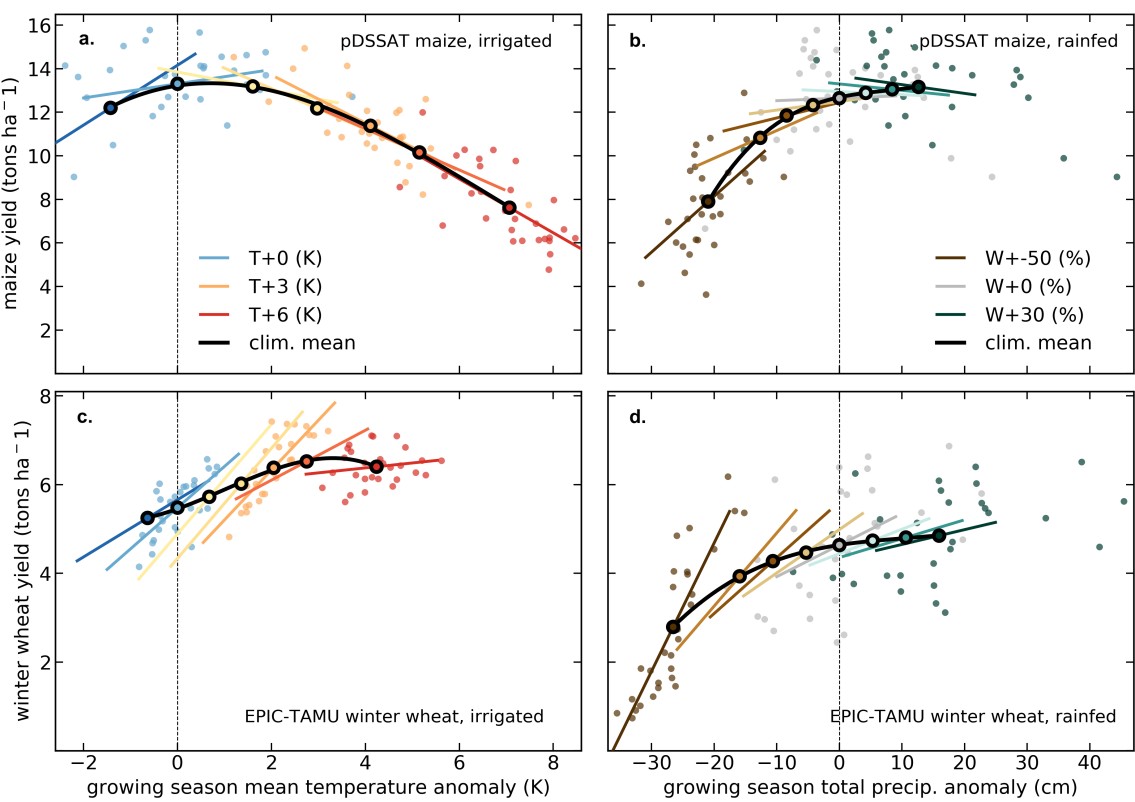

**Figure 1.** Example showing distinction between crop yield responses to year-to-year and climatological mean shifts in climate variables, showing representative high-yield regions for maize in pDSSAT (northern Iowa, top row) and winter wheat in EPIC-TAMU (France, bottom row). Left column (**a & c**) shows irrigated crops, all temperature cases with other variables held at baseline values, and right column (**b & d**) shows rainfed crops, all precipitation cases. Figure shows A0 output, in which growing seasons shift under future climate, so local growing-season temperature changes can differ from prescribed uniform offsets: for example, a 6 K applied uniform warming results in a growing season temperature warmer by ∼7 K for maize in Iowa (top right) but by less than 6 K for wheat in France (bottom right). Open black circles mark climatological mean yields and bold black lines show a third-order polynomial fit through them. Colored lines show linear regressions (by orthogonal distance regression) through the 30 annual yields of each parameter case. Colored circles show annual yields for selected cases. Differences in slopes of colored and black lines mean that responses to year-to-year fluctuations differ from those to longer-term climate shifts. Differences are generally stronger for wheat (bottom) than maize (top). Note that for rain-fed crops, slope differences in this representation could also result from correlated precipitation and temperature fluctuations in the baseline timeseries, but P-T correlations do not contribute to the effects shown here. Such correlations would complicate emulations based on year-to-year yields but would not necessarily bias them.

A second factor of importance is that any nonlinearity in crop responses will itself lead to a distinction between climatological and year-to-year fits, even if distributional differences are negligible. Given the interannual variations in the climate timeseries, the mean annual yield response to a perturbation is not the same as the response of the climatological mean yield.

The effect of nonlinearity may be particularly relevant for precipitation since model crop yields drop steeply and nonlinearly with increasing dryness. (Crop yields should drop under excess precipitation as well, but process-based models do not capture losses in saturated conditions well (Glotter et al., 2015; Li et al., 2019).)

In the GGCMI Phase 2 experiment, the imposed perturbations involve no changes in underlying distributions. The choice is reasonable since climate models do not agree on distributional changes. Most models do project small mean increases in growing-season temperature variability in cultivated areas and can produce substantial local changes, but models disagree on spatial patterns. For example, in models of the Coupled Model Intercomparison Project Phase 5 (CMIP5) archive, in the the high-end RCP (Representative Concentration Pathway) 8.5 climate projections to the year 2100 (Riahi et al., 2011), growing season daily maximum temperature variability over currently cultivated rice areas (weighted by production) increases by 10% in HadGEM2-ES but only by 0.4% in MIROC-ESM-CHEM. (See Supplemental Section S2.) We therefore explicitly test the assumption that distributional changes are not consequential for climatological mean yields: in Section 4.3, we confirm that an emulator trained on the GGCMI Phase 2 dataset can successfully reproduce yield changes under a full climate model projection.

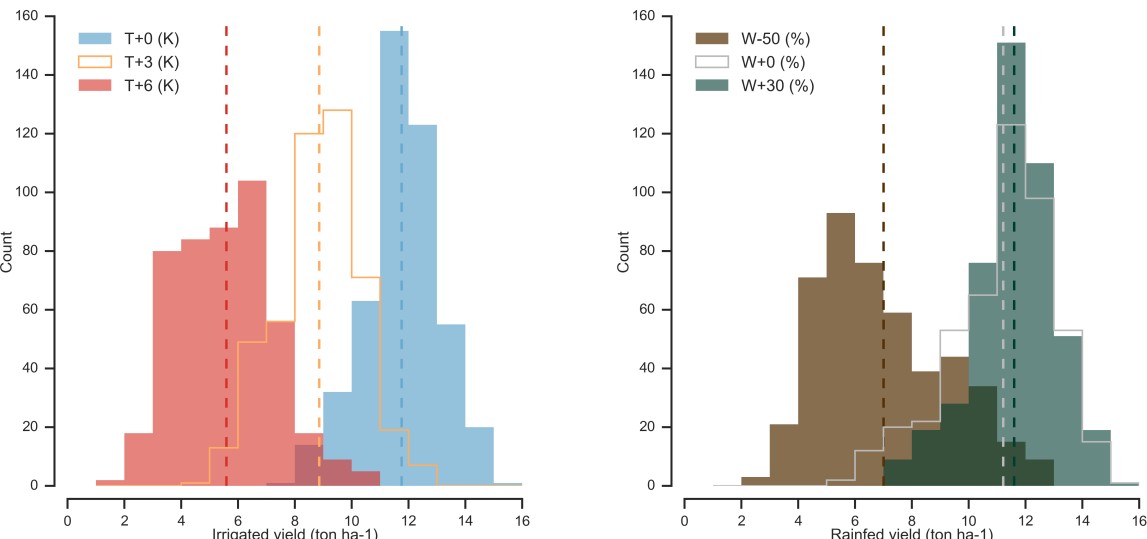

**Figure 2.** Example showing results of increased crop-yield sensitivity to year-to-year climate variations under climate stress. Yield distributions are from examples of Figure 1, top row, of maize in Iowa, (**left**) for irrigated maize in scenarios of altered temperature and (**right**) for rainfed maize in scenarios of altered precipitation. Because yield sensitivities rise under strong warming or drying, distributions of year-to-year crop yields widen in T+6 and P-50% scenarios relative to present-day simulations, even though all input climate timeseries have identical variance for temperature. Note: precipitation changes have different variance since the perturbations are fractional.

Note that even though distributions of climate variables are unchanged in the GGCMI Phase 2 simulations, the spread in annual yields still becomes wider in highly impacted climate states, because of the nonlinearity of yield responses (Figure 2). In the GGCMI Phase 2 dataset, all crops except rice show greater year-to-year yield variance in conditions of extreme

climate stress. (Rice is typically irrigated and experiences no water stress in simulations.) Increased variance has been noted in previous studies. For example, Urban et al. (2012) used statistical models trained on present-day yields to find a projected future increase in yield variance of U.S. maize of 20% per degree K temperature rise. Although the authors do not diagnose a specific cause of that increase, they discuss multiple potential mechanisms, including nonlinearity in responses.

## 3 Emulation

Emulation involves fitting individual regression models from GGCMI Phase 2 output for each crop and model and 0.5 degree geographic pixel; the regressors are the applied perturbations in $CO_2$, temperature, water, and nitrogen (CTWN). Here, we largely discuss emulations of climatological mean crop yields with no growing season adaptation (A0 scenarios) but note that any output of the crop models can potentially be emulated. We provide separate emulations of irrigated and rainfed yields and applied irrigation water (pirrww in mm $yr^{-1}$) in both A0 and A1 scenarios, meaning that each model and crop combination results in 6 sets of regressions. See Supplemental Material Sections 3, 4, and 6 for these additional emulation cases.

### 3.1 Statistical model

For the statistical model of crop yields as a function of CTWN, we choose a relatively simple parametric model with a third-order polynomial basis function (Equation 1). If the climatological mean response is relatively smooth, then a simpler form provides a reasonable fit that allows for some interpretation of resultant parameter weights. A relatively simple parametric form also allows fast model emulation at the grid-cell level, rather than requiring spatial aggregation. Emulating at the grid-cell level preserves the spatial resolution of the parent models and means that emulators indirectly include any yield response to geographically distributed factors such as soil type, insolation, and the baseline climate.

The third-order polynomial CTWN model of Equation 1 contains 34 terms, since the $N^3$ term is omitted, as it cannot be fitted in a training set sampling only three nitrogen levels. To facilitate comparing emulators parameter by parameter, we hold this functional form across locations, crops, and models, except for several necessary distinctions: regressions for irrigated crops do not contain W terms, and regressions for models that do not sample the nitrogen levels omit the N terms. Results shown throughout the paper use this full specification, but we also show (in Section 3.2 below) that for all but two models, 11 terms can be dropped without significant reduction in emulator fidelity. The higher specification of the 34-term model aids primarily in regions where crops are not currently grown. Most modeling groups submitted a sufficiently large training set that the 34-term model can be fit with standard ordinary least squares (OLS), but for models with lower sampling, it must be fit with a Bayesian Ridge regression method. (See Section 4 for evaluation of the fidelity of emulators constructed with Equation 1.)

$$
\begin{aligned}
Y \;=\; &K_1 \\
&+ K_2C + K_3T + K_4W + K_5N + K_6C^2 \\
&+ K_7CT + K_8CW + K_9CN + K_{10}T^2 + K_{11}TW \\
&+ K_{12}TN + K_{13}W^2 + K_{14}WN + K_{15}N^2 \\
&+ K_{16}C^3 + K_{17}C^2T + K_{18}C^2W + K_{19}C^2N \\
&+ K_{20}CT^2 + K_{21}CTW + K_{22}CTN + K_{23}CW^2 \\
&+ K_{24}CWN + K_{25}CN^2 + K_{26}T^3 + K_{27}T^2W \\
&+ K_{28}T^2N + K_{29}TW^2 + K_{30}TWN + K_{31}TN^2 \\
&+ K_{32}W^3 + K_{33}W^2N + K_{34}WN^2 + {\color{gray}K_*N^3}
\end{aligned}
\tag{1}
$$

In this study, we do not focus on comparing other functional forms or non-parametric models. In general, higher-order and interaction terms are expected to be important for representing crop yields. Higher-order terms are needed because crop yield responses to weather are well-documented to be nonlinear: e.g. Schlenker and Roberts (2009) for T perturbations and He et al. (2016) for W (precipitation). Interaction terms are needed since the yield response is expected to depend on interactions between the major inputs. For example, Lobell and Field (2007) and Tebaldi and Lobell (2008) showed that in real-world yields (with C and N fixed), the joint distribution in T and W is needed to explain observed yield variance. Other observation-based studies have shown the importance of the interaction between W and N (e.g. Aulakh and Malhi, 2005), and between N and C (Osaki et al., 1992; Nakamura et al., 1997). Some prior studies have used even more complex statistical specifications in crop model emulation: for example, Blanc and Sultan (2015) and Blanc (2017) use a 39-term fractional polynomial and "borrow information across space" by fitting grid points simultaneously across soil region in a panel regression. The GGCMI Phase 2 dataset allows fitting our simple third-order polynomial form independently at each grid cell while still providing a satisfactory emulation for all models and crops.

### 3.2 Feature importance and reduced statistical model

Because a simpler statistical model may improve the interpretability of its parameter weights, we also develop a reduced 23-term version that is satisfactory for most models and crops (Equation 2, with the 11 removed terms shown in gray). To identify terms that can be omitted, we apply a feature selection cross-validation process in which terms in the polynomial are tested for importance. Higher-order and interaction terms are successively added to the regression model, and in each case we calculate an aggregate mean absolute error (weighted by currently cultivated area) and eliminate those terms that do not contribute significantly to reducing error. The procedure is illustrated in Figure 3. We develop our reduced statistical model by considering yields over currently cultivated land in three models: two that provided the complete set of 672 rainfed simulations, i.e. without the $W_\infty$ simulations, (pDSSAT, EPIC-TAMU), and one that provided the smallest training set (121 input combinations, PEPIC). Although models exhibit different absolute levels of error, all three agree remarkably well on

feature importance, i.e. on which terms reduce error and which provide no predictive benefit. Agreement is indicated by
215 matching line slopes in Figure 3.

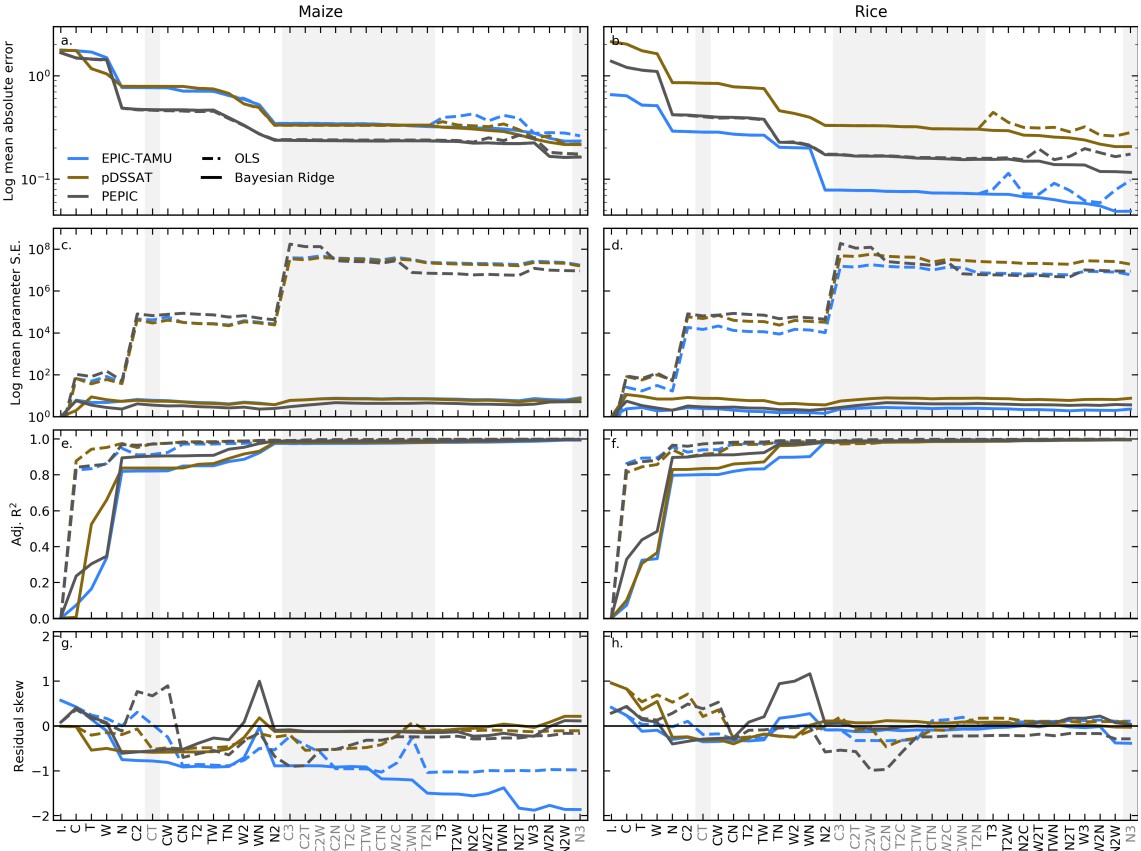

**Figure 3.** Illustration of results from the polynomial feature selection process for three different crop models (colors), for all grid cells with
more than 1,000 ha cultivated for maize (**left**) and rice (**right**). Solid lines are Bayesian Ridge regression results, and dashed lines are for
standard OLS. Rows show four metrics of fit quality, and x-axes show the terms successively tested in the statistical model, sequentially
added to the model in order from left to right. Terms that do not reduce the aggregate error are marked in gray and are not included in
the final model. **a & b:** log mean absolute error between emulated yield and simulated values calculated with a threefold cross-validation
process, where the emulator is trained on two-thirds of the data and predicts the remaining third. **c & d:** log mean standard parameter error.
The Bayesian Ridge method strongly reduces parameter error and results in more stable estimates. **e & f:** adjusted $R^2$ score for the fit at each
model specification. **g & h:** distribution of the residuals. Skewness is low at the high model specifications tested in all model cases other than
EPIC-TAMU maize.

$$
\begin{aligned}
Y = {} & K_1 \\
& + K_2 C + K_3 T + K_4 W + K_5 N + K_6 C^2 \\
& + K_a CT + K_7 CW + K_8 CN + K_9 T^2 + K_{10} TW \\
& + K_{11} TN + K_{12} W^2 + K_{13} WN + K_{14} N^2 \\
& + K_* C^3 + K_* C^2 T + K_* C^2 W + K_* C^2 N \\
& + K_* CT^2 + K_* CTW + K_* CTN + K_* CW^2 \\
& + K_* CWN + K_{15} CN^2 + K_{16} T^3 + K_{17} T^2 W \\
& + K_* T^2 N + K_{18} TW^2 + K_{19} TWN + K_{20} TN^2 \\
& + K_{21} W^3 + K_{22} W^2 N + K_{23} WN^2 + K_* N^3
\end{aligned}
\tag{2}
$$

The eliminated terms include many of those in C: the cubic; the CT, CTN, CTW, and CWN interaction terms; and all higher-order interaction terms in C. Finally, we eliminate one second-order interaction term in W and two in T. Implications of this choice include that nitrogen interactions are complex and important and that water interaction effects are more nonlinear than those in temperature. Note that some terms that did not reduce the aggregate error must still be included if a higher order version of that term provides benefit: for example, including the $T^3$ term requires also retaining $T^2$ and $T$ terms. The reduced-form emulator is acceptable across currently cultivated land for all model and crop combinations other than JULES soybean and spring wheat and PROMET soybean and rice. These cases involve yield responses that benefit strongly from inclusion of higher-order carbon dioxide interaction terms. Additional terms in the statistical model also help emulation in some geographic locations outside of currently cultivated regions, where yield responses are often non-standard. See Supplemental Material Section 7 for evaluation of the fidelity of emulators constructed with Equation 2 and for more details on JULES and PROMET.

## 3.3 Model fitting

To fit the parameters $K$, we use a Bayesian Ridge regularization method (MacKay, 1991) rather than ordinary least squares (OLS). The Bayesian Ridge method reduces volatility in parameter estimates when the sampling is sparse, by weighting parameter estimates towards zero, allowing the use of a consistent functional form across all models and locations. The choice slightly reduces mean absolute error for some of the high-order interaction terms in the model (Figure 3, top row) but drastically reduces standard parameter error in the model by stabilizing the estimates (Figure 3, third row). The estimation method scores relatively lower on adjusted $R^2$ for the simplest parameter specifications but quickly reaches parity with the OLS. We use adjusted $R^2$ as a metric because additional terms are penalized (Equation 3, where $n$ is the number of samples and $k$ is the number of features):

$$
R^2_{adj} = 1 - \frac{(n-1) \cdot (1 - R^2)}{n - k}
\tag{3}
$$

We use the implementation of the Bayesian Ridge estimator from the scikit-learn package in Python (Pedregosa et al., 2011).

An additional diagnostic of fit quality is the distribution of residuals: normally or near-normally distributed residuals imply that errors around the fit are random and unbiased. When fitting Equation 1 to the GGCMI Phase 2 dataset, the distribution of the residuals depends on the number of features included in the regression, the method for estimating the parameters, and the target distribution in the training set. The residuals are only normally distributed (pvalue > 0.05 in the Shapiro–Wilk test) for a
250 single model, PEPIC, for any specification tested here, but their skew is relatively small except in a single case, EPIC-TAMU maize (Figure 3, fourth row). While including higher-order terms in the statistical model generally reduces residual skew, for EPIC-TAMU maize it increases skew instead but also reduces the error in cross-validation, which we consider more important in the context of emulation. The residual distribution suggests that projections using the EPIC-TAMU maize emulator will tend to be biased high, but in practice, the overall magnitude of these errors is below 2% of yield changes. (See Section 4.2.)

## 4   Emulator evaluation

In this section we show illustrations of GGCMI model yield responses to climate perturbations and evaluate the ability of our emulators to reproduce them. Model emulation with the parametric method used here requires that crop-yield responses be sufficiently smooth and continuous to allow fitting with a relatively simple functional form; in Section 4.1 we show that this condition largely holds in the GGCMI Phase 2 simulations. In section 4.2 we evaluate metrics of emulator performance and
260 show that emulation errors – discrepancies between emulation and simulation – are generally small, especially when compared to the differences across crop models or to projected yield changes. We use the term *error* because, under the "perfect model" emulation approach, we take the simulation output to be perfect ground truth. We evaluate two separate error metrics, one more loose that incorporates information about the inter-model uncertainty, and one more stringent that tests out of sample prediction error within an individual model. For both metrics, emulation error is generally small other than in limited geographic locations,
usually where crops are not currently grown. Finally, in Section 4.3, we assess the emulator's ability to reproduce crop yields in a more realistic future simulation driven by a climate model projection, and find that its performance remains satisfactory. We analyze here results using the 34-term polynomial of Equation 1; see Supplemental Material Section 7 for analogous analysis of the 23-term polynomial of Equation 2.

### 4.1   Yield response

Crop yields show strong spatial differentiation across geographic regions, and emulators are able to readily reproduce these patterns. Figure 4 shows one example of simulated and emulated yields under current climate, using maize in LPJmL. Absolute emulation errors for this model-crop combination are low – 99.8% of grid cells have errors below 0.5 tons ha$^{-1}$ – but emulation errors as a percentage of baseline yield can be large in areas with low potential yield and no current cultivation in the real world (e.g. the Sahara, Patagonia). These regions are not currently viable for agriculture and may never become viable even under
extreme climate change. Emulator spatial skill varies across models and crops, with maize being the quantitatively easiest to emulate across all models and locations.

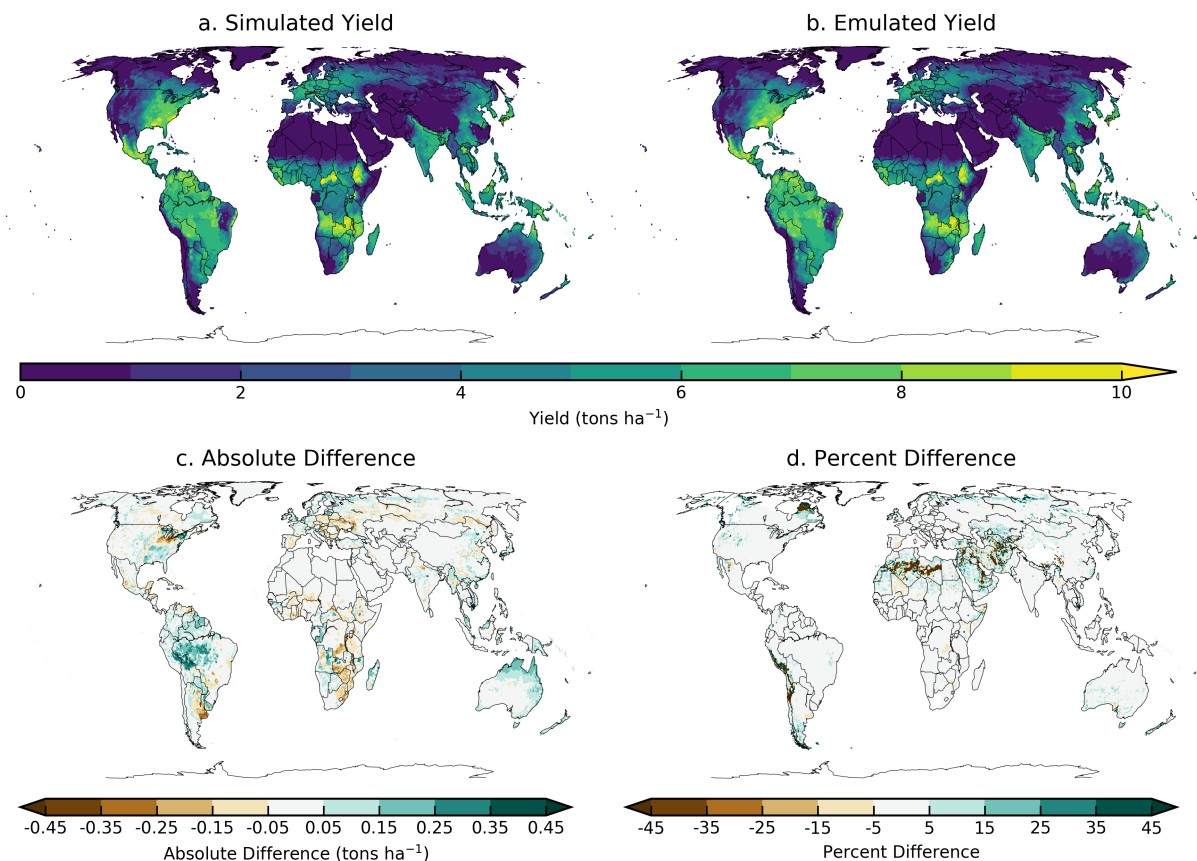

**Figure 4.** Illustration of spatial pattern in baseline yield successfully captured by the emulator: simulated (**a.**) and emulated (**b.**) yield under historical (1981-2010) conditions for rainfed maize from the LPJmL model. Absolute yield differences (**c.**) are less than 0.5 ton ha$^{-1}$ in almost all (99.8%) grid cells across the globe. Percent difference (from simulated baseline, **d.**) is below 5% in most (75%) grid cells currently cultivated in the real world. Approximately 7% of all grid cells, but only 3% of currently cultivated grid cell, have emulated yields that differ from the baseline simulation by more than 20%. Notable exceptions include areas with very low simulated baseline yield, including for example the Sahara, the Andes, and northern Quebec. Percent error weighted by cultivation area globally is essentially zero (see also Table 3). Performance varies by crop and model. See Supplemental Figures in Section 8 for more examples.

Yield responses to the four main drivers considered here (C, T, W, and N) are also quite diverse across locations, crops, and models, but in nearly all cases the local climatological mean responses are smooth enough to permit emulation with the functional form used here. Figure 5 illustrates the geographic diversity of responses within a single crop and model, for rainfed maize in pDSSAT. While the CO$_2$ responses (in ton ha$^{-1}$ ppm$^{-1}$) are quite similar, the precipitation response is stronger in more arid locations and the nitrogen responses appear strongly location-dependent. The heterogeneity in response supports the choice of emulating at the grid-cell level. In regions with current cultivation, yields evolve smoothly across the space sampled, and the polynomial fit captures the climatological-mean response to perturbations well. Emulators do perform poorly in a few

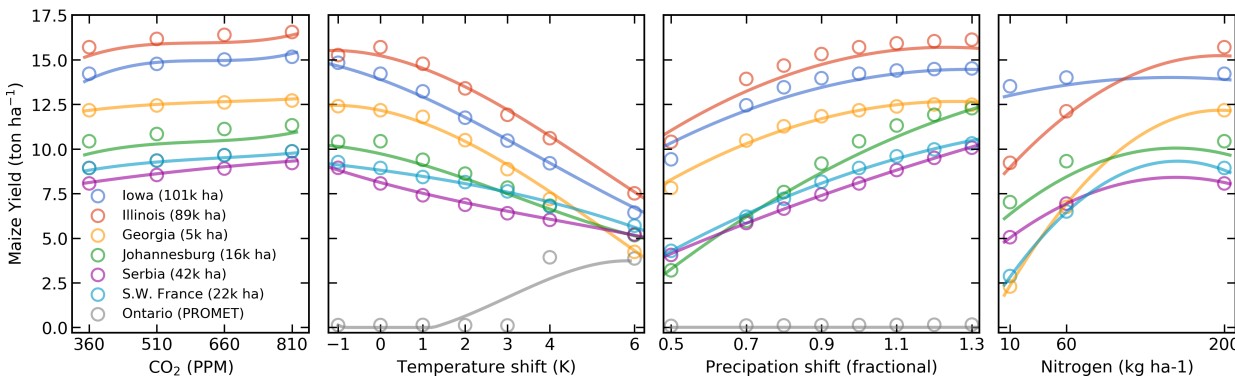

**Figure 5.** Illustration of spatial variations in yield response, which are successfully captured by the emulator. Panels show simulations (points) and emulations (lines) of rainfed maize in the pDSSAT model in six example locations selected to represent high-cultivation areas around the globe. Legend includes hectares cultivated in each selected grid cell. Each panel shows variation along a single variable, with others held at baseline values. Dots show climatological mean yields and lines the results of the full 4D emulator of Equation 1. In general the climatological response surface is sufficiently smooth that it can be represented within the sampled variable space by the simple polynomial used in this work. In some cases extrapolation would produce misleading results, and the emulator fails in conditions where yield response changes abruptly. Failure is illustrated here by rainfed maize in north-central Ontario for the PROMET model (in gray), which shows present-day yields of zero rising abruptly if temperature warms by 4 degrees.

regions that involve discontinuous or irregular yield responses. Poor performance is illustrated here with PROMET maize in northern Canada, which is too cold for maize at present in PROMET (0 ton ha$^{-1}$ yield) but shows an abrupt rise to moderate yields once temperature rises by 4 degrees. Under these conditions, the third-order polynomial cannot fit the response, and errors are high. See Section 4.2 for additional discussion.

Crop-yield responses in all models generally follow similar functional forms at any given location though with a spread in magnitude (Figure 6, which shows rainfed maize in northern Iowa in a selection of GGCMI models). Absolute yield differences between models can be substantial because some models are uncalibrated. In general, models are most similar in their responses to temperature perturbations and least similar to changes in $CO_2$. That is, $CO_2$ fertilization effects *within* a single model are consistent across locations, but $CO_2$ effects differ strongly *across* models.

Note that while the nitrogen dimension is important, it is also the most troublesome to emulate in the GGCMI Phase 2 experiment because of its limited sampling. The GGCMI Phase 2 protocol specified only three nitrogen levels (10, 60 and 200 kg N y$^{-1}$ ha$^{-1}$), so a third-order fit would be over-determined but a second-order fit can result in potentially non-physical results. Steep and nonlinear declines in yield with lower nitrogen levels mean that some regressions imply a peak in yield between the 100 and 200 kg N y$^{-1}$ ha$^{-1}$ levels (Figure 6, right). While reduced yields under high nitrogen levels are physically possible and could reflect over-application at particular times in the growing period, they are implausible at the magnitude shown here and likely an artifact of the fit. The Bayesian Ridge estimator mitigates the 'peak-decline effect' in the nitrogen dimension relative to ordinary least squares but does not entirely remove it. The polynomial fit also cannot capture the well-

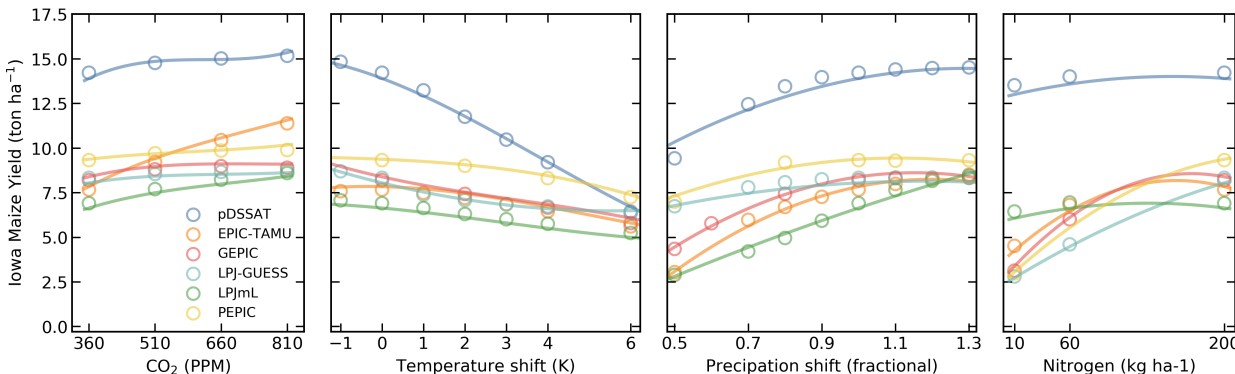

**Figure 6.** Illustration of variations in yield response across models, again successfully captured by the emulator. Panels show simulations and emulations from six representative GGCMI models for rainfed maize in the same Iowa grid cell shown in Figure 5, with the same plot conventions. Three models (PROMET, JULES, and CARAIB) that do not simulate the nitrogen dimension are omitted for clarity. Models are uncalibrated, producing spread in absolute yields. While most model responses can readily emulate with a simple polynomial, some response surfaces diverge slightly from the polynomial form, producing emulation error (e.g. pDSSAT here, for water), but resulting error generally remains small relative to differences across models.

documented saturation effect of nitrogen application (e.g. Ingestad, 1977) as accurately as would be possible with a non-parametric model.

## 4.2 Emulator performance metrics

Our emulators collectively consist of nearly 3 million individual regressions, so developing concise performance metrics poses a challenge. No general agreed-upon criteria exist for defining an acceptable crop model emulator, so we present two different metrics below, one relatively loose and one more stringent. Both metrics assess the ability of the emulator to reproduce simulated crop yields in the GGCMI Phase 2 experiment. In this section we show only results from emulators based on the 34-term Equation 1; see Supplemental Material Section 7 for analogous assessment of emulators based on the 23-term Equation 2.

*1. Normalized error.* We take as our first metric what we term the "normalized error", which compares the fidelity of an emulator to the inter-model spread. For a multi-model comparison exercise like GGCMI Phase 2, a reasonable though loose emulator criterion is that its errors be small relative to inter-model differences. The normalized error $e$ is defined separately for each C,T,W,N scenario $s$ as the difference between emulated and simulated fractional yield changes, normalized by the standard deviation in simulated changes across all models:

$$e_s = \frac{F_{em,s} - F_{sim,s}}{\sigma_{sim,s}} \qquad (4)$$

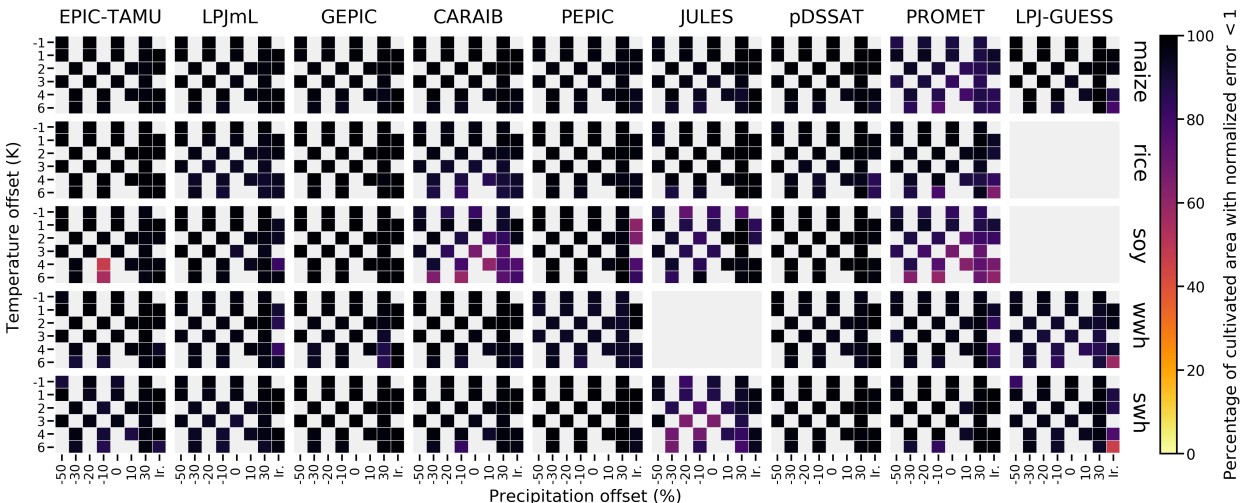

**Figure 7.** Assessment of emulator performance over currently cultivated areas based on normalized error (Equations 5). We show performance of all 9 models emulated, over all crops and all sampled T and W inputs ("ir." indicates the irrigated $W_\infty$ setting) but with $CO_2$ and nitrogen held fixed at baseline values. Large columns are crops, and large rows are models; squares within are T, W scenario pairs. Colors denote the fraction of currently cultivated hectares ("area frac") for each crop with normalized area $e$ less than 1 indicating the error between the emulation and simulation less than one standard deviation of the ensemble simulation spread. Of the possible 63 scenarios at a single $CO_2$ and N value, we consider only those for which all 9 (8 for rice, soybean, and winter wheat) models submitted data (Figure S1) so the model ensemble standard deviation can be calculated uniformly in each case. JULES did not simulate winter wheat and LPJ-GUESS did not simulate rice and soybean. Emulator performance is generally satisfactory, with some exceptions. Emulator failures (significant areas of poor performance) occur for individual crop-model combinations, with performance generally degrading for colder and wetter scenarios.

where $F$ is the fractional change in yields $Y$ between scenario $s$ and baseline $b$:

$$F_s = \frac{Y_s - Y_b}{Y_b} \qquad (5)$$

We calculate the mean error for each grid cell, model, and crop in each C,T,W,N scenario by comparing emulated and simulated yields. A normalized error $e < 1$ means that any deviation of the emulation from the simulation is less than one standard deviation of the inter-model spread.

Evaluation of this metric implies that GGCMI Phase 2 emulators are generally satisfactory. Emulator performance is il-
320 lustrated in Figure 7, which shows all models and crops crops over currently cultivated area. Over all crops and models, the average normalized error $e < 1$ over 95% of currently cultivated area. For maize, the most tractable crop to emulate, all 9 models return $e < 1$ over 97% of currently cultivated area. Only three crop-model combinations are problematic, returning $e < 1$ over less than 90% of cultivated area even when using the 34-term statistical model: PROMET and CARAIB for soybeans (79% and 83%), and JULES for spring wheat (85%). Misfits typically occur when models show strong discontinuities in yield
response (as shown in Figure 5), or when carbon dioxide fertilization gains interact nonlinearly with changes in temperature

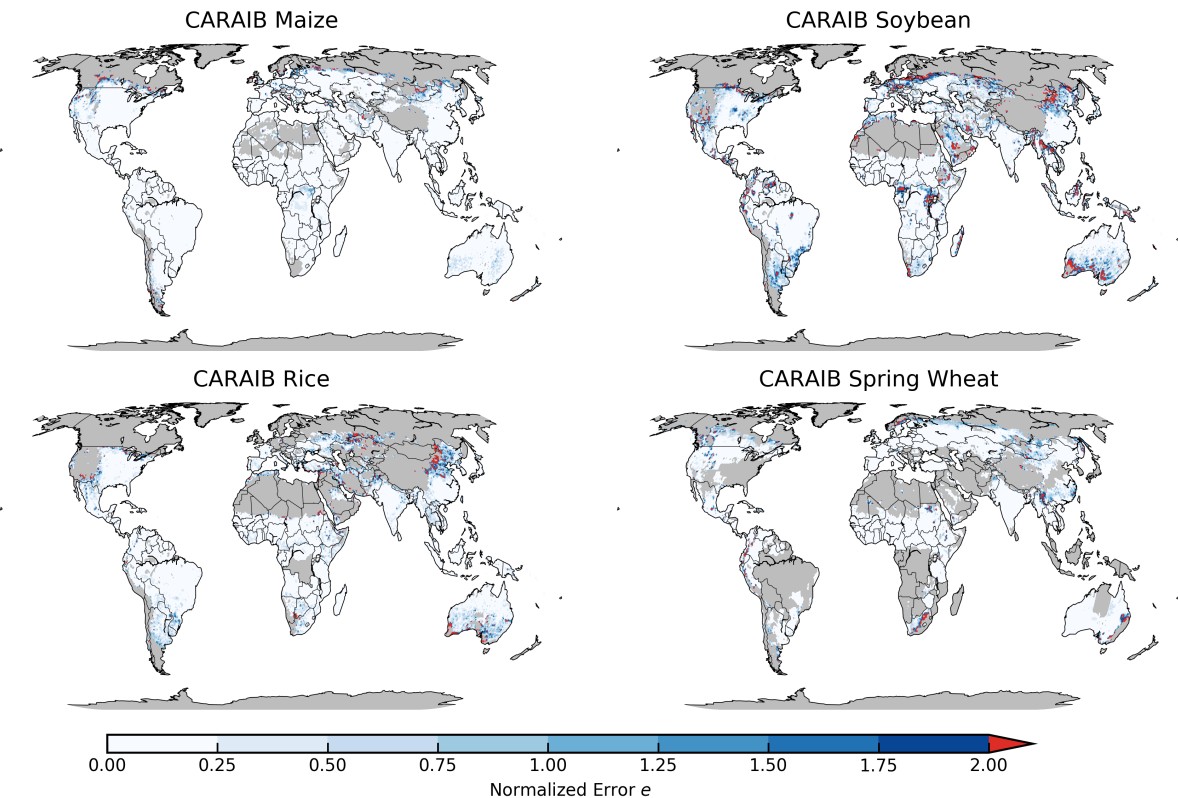

**Figure 8.** Illustration of our first test of emulator performance, applied to the CARAIB model for the T+4 scenario for rainfed crops. Colors indicate the normalized emulator error $e$, where $e > 1$ means that emulator error exceeds the multi-model standard deviation. For consistency, we show $e$ only for geographic areas simulated by at least six models and where baseline yields are greater than 0.5 ton ha$^{-1}$. Emulator performance is generally good relative to model spread in areas where crops are currently cultivated (compare to Figure S2-S3) and in temperate zones in general; emulation issues occur primarily in marginal areas with low-yield potentials.

or water. Including higher-order C terms helps in the latter case but does not reduce emulator errors to zero. See Supplemental Figures S22-S23 for examples of worst-case emulator failures.

    While Figure 7 shows only currently cultivated land, performance can be worse in locations where crops are not currently cultivated or on marginal lands where current potential yields are low. In general, emulator performance is poor anywhere that models show steep yield changes once some threshold has been reached. Some of these cases involve complete crop failures in a changed climate, but most involve yield improvements: abrupt gains in regions that are too cold or dry under current conditions but that become viable given warming or wetting. Figure 8 illustrates this effect for CARAIB in the T+4 scenario, showing normalized error over all simulated areas with non-zero baseline yield and at least six models providing simulations. CARAIB emulator performance is generally good where crops are grown but can be poor ($e > 2$) in arid or mountainous zones, e.g. the edges of the Sahara, inner Mongolia, South Africa and southern Australia. Effects will vary by crop model as they differ in process implementations; see the different model description papers referenced in Table 1 for more details. Note that the

choice of statistical model for emulation involves a trade-off in the spatial pattern of errors. The 34-term statistical model used here maximally improves emulator fidelity in problematic "fringe" areas at the expense of lowering it slightly over high-yield areas. For example, over currently cultivated land, CARAIB maize emulators have normalized error $e < 1$ over 98.5% of area with the full 34-term Equation 1 but over 98.8% with the reduced 23-term Equation 2. The effect is reversed over uncultivated land, with CARAIB maize emulators showing $e < 1$ over 93.7% of area with the full Equation 1 but only over 88.7% of area with the reduced Equation 2.

The normalized error assessment is relatively forgiving for several reasons. First, it is an in-sample validation, with the emulation evaluated against the simulations actually used to train the emulator. Had we used a spline interpolation, the error would necessarily be zero. Second, the metric scales emulator fidelity not by the magnitude of yield changes in the evaluated model but by the spread in yield changes across models. The normalized error $e$ for a given model then depends on the particular suite of models considered in the intercomparison exercise. The rationale for the choice is to relate the fidelity of the emulation to the true uncertainty, which we take as the multi-model spread, but the metric then has the property that where models differ more widely, the standard for emulators becomes less stringent and vice versa. In GGCMI Phase 2 the effect is manifested in the higher normalized errors for soybeans across all models, which result not because soybean yields are difficult to emulate but because models agree more closely on yield changes for soybeans than for the other crops.

*2. Out-of-sample validation.* We provide a second, more stringent test of emulator performance via a threefold cross validation (also termed an out-of-sample validation). In this test the GGCMI Phase 2 dataset is split randomly into two parts, with 90% of the data used to train (calibrate) the model and the held-out 10% used to test (evaluate) the fidelity of the resulting emulator. The procedure is repeated three times; in each case we calculate the root mean square error (RMSE) between the emulated (predicted) and actual simulated test set values, and then average the three results. The result is a single metric for each grid cell for each model-crop combination. As a last step, we normalize the error metric for each grid cell by dividing by its maximum yield change over the entire CTWN dataset. (Since all models have submitted the extreme T+6 scenario, this normalization choice is not problematic.) Note that this validation exercise is independent of the procedure for generating the final published emulator values, which are generated using the full CTWN dataset.

The resulting error metric is generally low. Table 3 shows the yield-change-normalized RMSE for rainfed crops in all models over currently cultivated land, both in selected major producing regions and in the global average. We include all simulations in the CTWN space and report the average error value in Table 3. Global mean grid-cell RMSE is below 5% of maximum yield changes in all cases, or in absolute terms less than 0.2 ton ha$^{-1}$ for all except JULES soybean simulations (0.36 ton ha$^{-1}$). Emulators for rainfed and irrigated crops have similar fractional errors, but since irrigated crops experience lower yield changes across the CTWN scenarios, they also have lower absolute errors. See Supplemental Material Section 9 for maps of cross validation RMSE for each crop and model.

Note that this relatively simple metric may be over-conservative. The randomized sampling protocol for dividing training and test sets can mean that a training set omits edge simulations at the highest or lowest value in CTWN space. The test prediction then involves extrapolating out of the training set range (e.g. predicting a T+6 case when the training set extends only to T+4), an improper use of an emulator. RMSE values would be lower if we had used a more careful sampling strategy

**Table 3.** RMSE of emulator replication of simulated yields of rainfed crops, stated as a percentage of simulated yield change. Values are the mean grid cell error as a percentage of simulated yield change, over all currently cultivated grid cells weighted by cultivation area, for selected major regions (NA: North America, SA: South America). For comparison, global mean values are shown in parentheses. Errors are calculated using the 90-10 cross validation scheme described in text, with the model trained on 90% of the data and validated on the held-out 10% (repeated twice). All fits are made with the Bayesian Ridge method; for context, we mark with * those cases where the Bayesian Ridge is required because the OLS linear model fails (e.g. PEPIC, which has the lowest number of samples at n=121).

| Model | NA Maize | SA Soybean | SE Asian Rice | NA S. Wheat | European W. Wheat |
|---|---|---|---|---|---|
| **CARAIB** | 0.7 (0.9) | 2.4 (2.4) | 2.4 (2.4) | 1.3 (1.4) | 2.7 (1.9) |
| **EPIC-TAMU** | 2.4 (1.8) | 1.8 (2.6) | 1.6 (1.6) | 1.8 (1.9)* | 1.1 (1.1) |
| **JULES** | 2.6 (2.6) | 4.6 (4.0) | 1.6 (1.7) | 2.0 (2.2) | NA |
| **GEPIC** | 2.1 (2.4) | 1.0 (1.2) | 2.0 (2.1) | 3.7 (3.3) | 4.0 (2.9) |
| **LPJ-GUESS** | 1.0 (1.1) | NA | NA | 1.0 (1.3) | 1.0 (1.2) |
| **LPJmL** | 1.8 (1.8) | 1.1 (1.3) | 1.2 (1.1) | 0.8 (1.1) | 1.5 (1.3) |
| **pDSSAT** | 1.9 (1.7) | 1.2 (1.1) | 1.7 (1.6) | 1.1 (1.3) | 1.4 (1.5) |
| **PROMET** | 3.4 (2.7)* | 2.0 (2.7)* | 2.1 (1.8)* | 4.3 (3.7)* | 4.6 (3.4)* |
| **PEPIC** | 1.8 (1.8)* | 1.4 (1.9)* | 1.4 (1.4)* | 2.3 (2.3)* | 4.9 (2.9)* |

that precluded extrapolation (e.g. "leave-one-out"). For additional discussion of more detailed potential evaluation metrics, see e.g. Castruccio et al. (2014).

## 4.3 Emulation of realistic climate projections

Finally, we test the ability of an emulator based on the GGCMI Phase 2 perturbed mean training set to reproduce the response of a crop model driven by a realistic, evolving climate scenario. Our emulators are trained only on growing-season means, and the GGCMI Phase 2 exercise involved only changes in means. We therefore seek to assess whether changes in the higher moments of temperature and precipitation distributions in a climate projection might have effects that lead to significant emulator error. Note that we are not asking whether year-to-year climate variability matters to crop yields; this point is well-established (Ray
et al., 2015). The question instead is whether a realistic future climate projection involves *changes* in variability large enough that they compromise an emulator based on the GGCMI Phase 2 dataset.

To assess this potential error, we generate new crop model simulations using the LPJmL crop model (taken as a representative of GGCMI models), driven by a climate simulation from the the Coupled Model Intercomparison Project Phase 5 (CMIP5) archive (Jones et al., 2011; Martin et al., 2011; Taylor et al., 2012). To maximize any potential bias, we choose a climate model
(HadGEM2-ES) that exhibits relatively large changes in growing-season temperature variability among CMIP5 members (Supplemental Table S1), and use the high-end RCP 8.5 scenario. We also hold $CO_2$ fixed to emphasize the results of temperature and precipitation changes, in the absence of the beneficial effects of increased $CO_2$. We then compare the resulting simulated yields to the output of the GGCMI LPJmL emulator driven by the HadGEM2-ES yearly growing-season T and P anomalies

(Figure 9). The GGCMI LPJmL emulator is able to capture the yield changes well: for all crops, emulated and simulated global
production in the last decade of the simulation are identical to within 1.5%. These results imply that globally, the results of
future distributional shifts on climatological yields are small relative to the effects of mean changes (Figure 9). The GGCMI
LPJmL emulators also reproduce decadal variations in yields, which are especially strong in spring wheat grown in northern
latitudes (Figure 9, right) and even capture much of the residual year-to-year yield variability: $R^2$ of emulated vs. simulated
annual yield anomalies relative to the 10-year running mean is 0.8 for spring wheat (and $\sim$0.3 for all other crops).

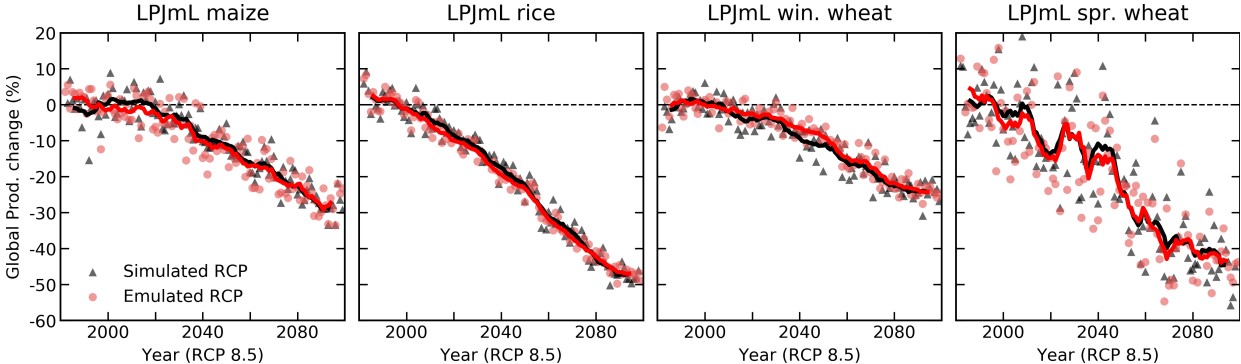

**Figure 9.** Test of emulator performance in reproducing yield simulations made with a realistic climate projection. Panels show simulated
(black) and emulated (red) global production for four crops from the LPJmL model, driven with temperature and precipitation outputs from
the HadGEM2-ES climate model for the RCP8.5 scenario. In both cases nitrogen and $CO_2$ are held fixed, at 200 kg ha$^{-1}$ and 360 ppm.
Points show yearly global production change from the 1981-2010 baseline, and lines show a 10-year running mean. See text for discussion of
relating the HadGEM2-ES temperature timeseries to the appropriate offset used in emulation. Emulators trained on uniform climatological
offsets reproduce well the simulated production response under a realistic climate scenario: yields at end of century match to within 1.5%.

Distributional effects might be expected to be stronger at high latitudes because temperature and precipitation variability
are larger there, so changes in variability can be correspondingly more important. However, we find that most crops (spring
wheat, winter wheat, and maize) show no emulator bias that grows with latitude. Rice is the exception: the climatological-
mean emulator slightly over-predicts yield losses in the tropics and under-predicts losses at higher latitudes (where little rice
is currently grown). Poleward of 30 degrees latitude, the LPJmL simulation under the HadGEM2 RCP scenario shows a 49%
reduction in rice yields by end-of-century (without growing-season adaptation), but the GGCMI-based emulator produces a
reduction of only 39% (Supplemental Figure S11). These losses are concentrated in the lower mid-latitudes: only 21% of global
rice is cultivated poleward of 30 degrees and only 1% poleward of 45 degrees.

It is worth noting two complications involved in comparing emulated to simulated yields under a realistic climate change
scenario, as in Figure 9. First, it is not trivial to choose how to relate temperature or precipitation in the evolving climate scenario
to the $T$ and $P$ offsets used as regressors to the emulator. Using growing-season mean temperature can lead to complications
if crop models assume that growing season lengths shift under climate change. For consistency, we match the temperature
changes in the climate scenario to their equivalent emulator regressors by calculating means over the fixed baseline growing

season. This choice ensures that the emulation is appropriately matched to the simulation. Second, although the emulator outputs an estimated yield change, the baseline from which that yield change is calculated will be different between simulation and emulation because the historical climate timeseries are not identical. For example, the baseline (1981-2010) yield of winter wheat simulated by LPJmL using the AgMERRA timeseries as part of GGCMI Phase 2 is 7% lower than that simulated using the HadGEM2-ES timeseries. To minimize the effects of different historical climate assumptions, we drive the emulator with the anomaly of the climate scenario from its own 1981-2010 mean. Bias in the historical climate timeseries could in theory produce discrepancies between emulated and simulated yield changes because of the nonlinearities discussed in Section 2.2, but the effect appears to play little role in the LPJmL comparison of Figure 9.

## 5  Emulator results and products

The crop model emulators developed here can be used for a variety of applications because the emulator transforms the discrete simulation samples into a continuous response surface at any geographic scale. One use is construction of continuous agricultural damage functions in a flexible format. As an example, we present in Figure 10 global damage functions over each of the four dimensions tested in this study, constructed from the 4D emulation of each crop model.

These damage functions are useful in diagnosing commonalities and differences in the responses of crop models. In most cases, models agree on the sign of responses to individual factors, but the spread in model responses is comparable to the median response. Inter-model spreads are largest for spring wheat and smallest for soybeans, as also shown in Figure 7. Model responses to individual factors conform to expectations. As expected, the $CO_2$ response is smallest for maize, which is a C4 crop, and the nitrogen response is smallest for soybeans, which are efficient fixers of atmospheric nitrogen. Nitrogen responses in crops other than soybeans are relatively similar, and most models show saturation beginning at values less than 200 kg ha$^{-1}$. In nearly all crop models and for all crops except spring wheat, damages from reduced precipitation exceed benefits from increased precipitation. Spring wheat is the exception, likely because it is grown in high latitudes where rainfall may be limiting. Rice, by contrast, which is generally grown in locations with abundant water, shows nearly no benefit from increased precipitation. Note that these damage functions do not consider whether increased precipitation might permit cultivation in new areas, and also that crop models generally do represent damages from excess soil moisture well (Li et al., 2019).

The GGCMI Phase 2 emulators are also intended as a tool for impacts assessments. The T and W functions presented in Figure 10 are not true global projections because they emulate the consequences of uniform shifts across the globe. However, the emulator allows building analogous damage functions based on climate model output, which has more realistic spatial patterns of changes in temperature and precipitation. In Figure 11, we show emulated maize responses for three crop models under the RCP8.5 scenario, using output from three climate models from the CMIP5 archive. Losses are shown as a function of mean growing-season temperature over currently cultivated land. While these damages functions aggregate over all currently cultivated land, the global coverage of GGCMI Phase 2 allows impacts modelers to develop damage functions for any desired geopolitical or geographic region larger than 0.5 degrees in latitude and longitude.

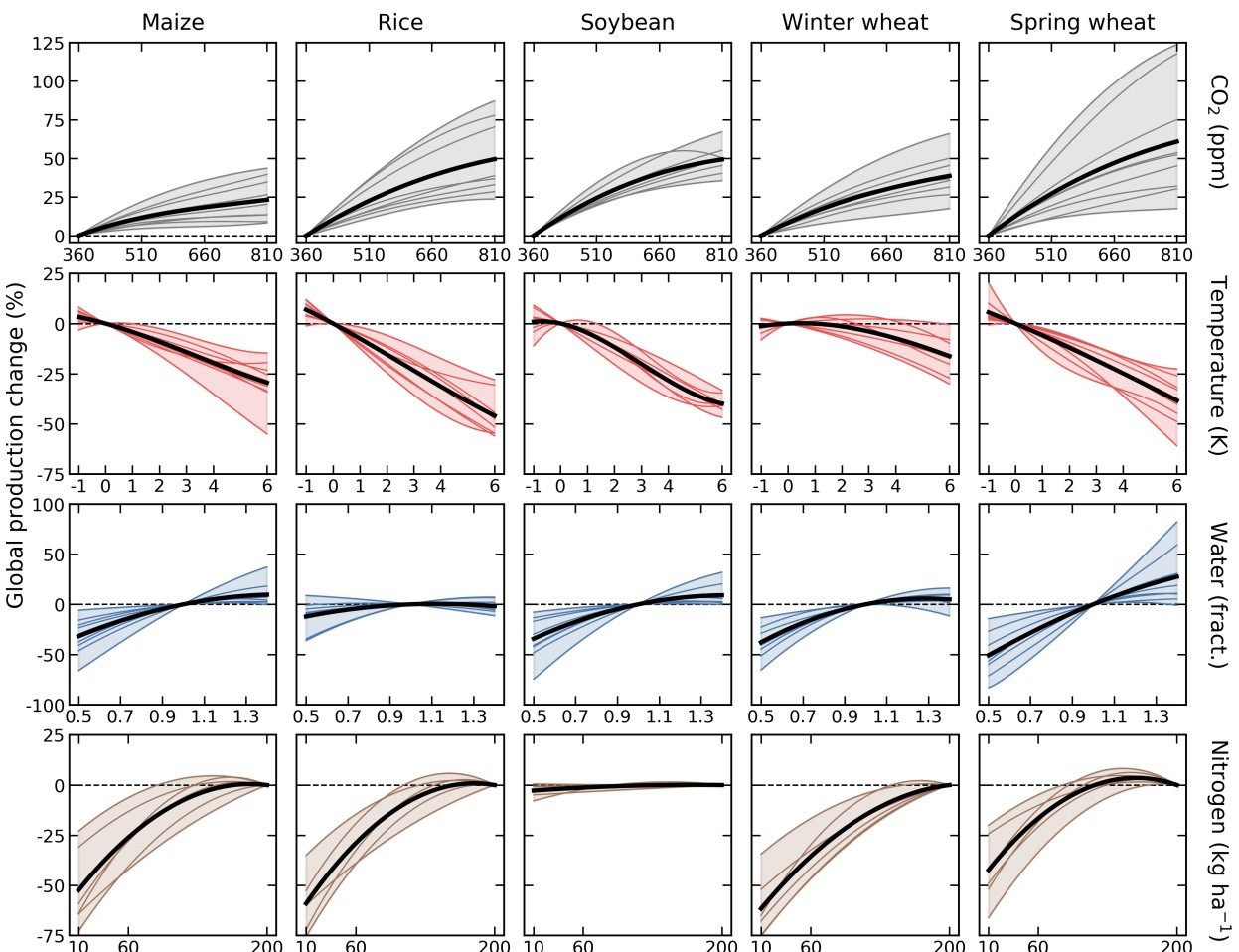

**Figure 10.** Emulated global damage functions for the five crops over the four CTWN dimensions varied in GGCMI Phase 2. Black line shows the multi-model mean and shaded area, and colored lines show the individual models. The number of models in each case varies because some models did not provide all crops or simulate the N dimension. Each panel shows response to one covariate for rainfed crops, with all others held constant at baseline values (e.g. C = 360 ppm, N = 200 kg ha$^{-1}$). Damages are reported as percent change in global production over currently cultivated land relative to the 1981-2010 baseline. Note that y-axis ranges are not uniform. As expected, the N response is smallest in soybeans, which are nitrogen fixers, and the C response is smallest in maize, a C4 crop. See Supplemental Figure S12 for an analogous figure identifying each crop model and Supplemental Figure S13 for damage functions for the A1 (adaptive growing season) emulators, which have reduced temperature responses.

The emulated responses of Figure 11 allow diagnosing the factors of greatest importance to projected yield changes under future climate change. In the maize example here, temperature is the overwhelmingly dominant factor for pDSSAT, but $CO_2$ responses are far larger in PROMET. $CO_2$ is important across models for spring wheat, see Figure S14. For all crop models, the aggregated effects of precipitation changes are negative, exacerbating yield losses (compare T and T+W cases) because

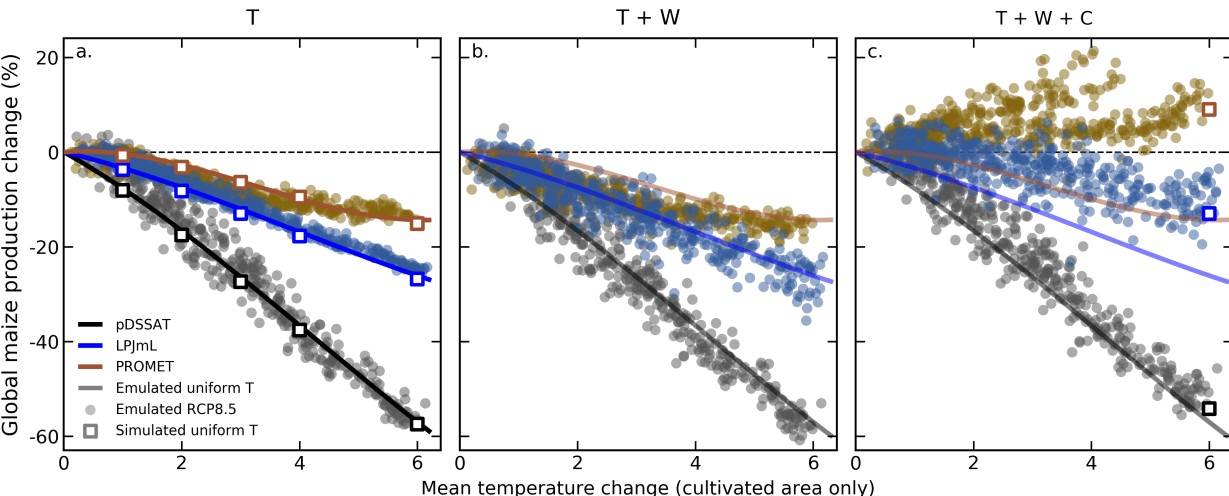

**Figure 11.** Illustration of the use of the emulator to study the factors affecting yields in a more realistic climate scenario. Figure shows emulated yield changes (relative to 1981-2010) for maize (both rainfed and irrigated) on currently cultivated land under RCP8.5 climate projections from three representative CMIP5 climate models ((HadGEM2-ES, GFDL-ESM2M, and IPSL-CM5A-LR), using changes to T only (**a**), to T and W (**b**), and to T, W, and C (**c**). The x-axis is the mean growing-season temperature change over cultivated land, computed using the historical growing season; note that these values will be higher than the corresponding global mean temperature change. Dots are emulated yearly global production changes to 2100 (90 years × 5 climate timeseries = 450 per crop model), with x-axis the mean historical growing-season T shift over all grid cells where maize is grown (unweighted by within-cell cultivated area). **a:** Using only temperature changes allows comparing regional simulated and emulated values. Open squares are GGCMI Phase 2 simulated values for each T level, with CWN held at baseline; bold lines are emulated values over uniform $\Delta$T shifts (repeated in each panel). Emulation uncertainty (compare squares to lines) is small relative to differences across climate and crop models, and mean yield changes are similar whether T changes are applied as a uniform shift or in a more realistic spatial pattern (compare lines to dots). **b:** Adding in precipitation changes increases yield spread across climate projections and depresses yield slightly. No squares are shown in **b** because the GGCMI uniform offsets of both T and W are not directly comparable to GCM-specific changes of T and W in a climate projection. **c:** $CO_2$ fertilization is small in pDSSAT, moderate in LPJmL, and very large in PROMET. The separation of groups of points in PROMET (gold) results because CMIP5 climate sensitivities differ by nearly a factor of two; points at far right are under the highest-sensitivity model, HadGEM2-ES. In RCP8.5, the 30-year-average $CO_2$ at end of century is 807 ppm (Riahi et al., 2011). For comparison, open squares in **c** show GGCMI-2 simulated production changes at T+6, W=0, C=810 ppm. (Note that in these climate projections, mean $CO_2$ levels when T > 5.8 degrees is 912 ppm.) See Supplemental Figures S14-15 for analogous figures for other crops (spring wheat and soybeans).

precipitation in HadGEM2 actually declines over maize cultivation regions, especially in Central and S. America. Precipitation
effects are relatively small, however, as manifested in two ways: as only a small mean shift in yield projections for individual crop models (compare T and T+W cases), and as a relatively small increase in the spread of points here at a given temperature, despite the fact that the climate projections used involve different relationships between temperature and precipitation change. By contrast, the carbon dioxide fertilization response for PROMET is so large that projections from climate models of different

sensitivities ($\Delta T/\Delta CO_2$) become clearly separated in Figure 11. PROMET yield responses would be more similar if plotted as a function of $CO_2$ than they are when plotted as in Figure 11 as a function of temperature change.

Disaggregating the factors driving crop yield changes also highlights the fact that errors of emulation are much smaller than the spread across crop models or even across different climate simulations. PROMET is the most quantitatively difficult model to emulate for maize, but its comparatively large emulation error (compare open squares to lines in T case) is still smaller than the spread simply due to different T patterns across climate simulations (Figure 11, left, compare differences between open squares and line with the spread in dots for a given temperature value). Uncertainties in the yield damage function due to projected patterns of temperature change are in turn smaller than spread due to differing model relationships of W and T changes (Figure 11, middle), and for PROMET are enormously outweighed by uncertainty in climate sensitivity (Figure 11, right). While emulator fidelity is important to ensure, it is important to recognize that these other uncertainties will dominate any impacts assessment exercise. Note that the pattern-related yield effects are actually relatively small for maize. (In Figure 11, left, compare lines, which show yield changes under uniform temperature shifts, to dots, which show changes under realistic warming scenarios). Pattern-related yield effects can be larger for other crops, and the uncertainties due to climate projection differences correspondingly larger: see for example soybeans in Supplemental Figure S15.

## 6 Discussion and conclusions

In this work we describe a new class of global gridded crop model emulators for five crops (maize, soybean, rice and spring and winter wheat) and nine process-based crop models, based on the GGCMI Phase 2 dataset, a set of crop model simulations run with systematic perturbations to carbon, temperature, precipitation, and nitrogen (CTWN). The goal of this project is to provide a lightweight tool that reproduces the output of large numerical simulations of process-based crop models. The resulting emulators should provide useful tools both for diagnosing crop model behavior and for climate impacts assessment, at least of large-scale time-averaged responses. Specific findings of this work include that:

- In crop models, the climatological mean yield responses to uniform perturbations in growing-season mean temperature and precipitation are very distinct from responses to historical weather fluctuations associated with the same mean differences. This result suggests that when emulating crop models, care must be taken if considering responses on both short and long timescales. The large GGCMI Phase 2 experiment allows us to emulate climatological-mean responses with a simple statistical model without relying on the "natural experiment" of year-to-year variations.

- Climatological mean responses in all models can be well-fit with a simple third-order polynomial in mean growing-season C, T, W, and N. The large GGCMI training set allows fitting in most cases with OLS, but use of a Bayesian Ridge regression provides additional stability and prevents overfitting. For most crop models, emulation is also possible with a simplified version of the statistical model with only 23 terms.

- The resulting emulators are highly flexible: they capture the strong geographic difference in crop yields and yields responses, can perform well on models with quite different sensitivities to climate or $CO_2$ changes. Emulators can

faithfully reproduce the output of process-based crop models in both in- and out-of-sample tests. Emulation error is generally small other than in localized regions where crops are not currently grown: across all models and scenarios, errors over currently cultivated land never exceed 5% of yield changes at either global or regional scale.

- Emulators trained on the GGCMI Phase 2 dataset, which samples over uniform climate perturbations, can effectively reproduce the behavior of crop models driven by realistic future projections of future T and P changes. This result suggests that any projected changes in weather distributions (temperature and precipitation variability) have relatively little effect on crop-model yield responses relative to changes in means, at least on the regionally aggregated level.

- The GGCMI emulators should provide powerful tools for both model comparison and impacts assessments. The emulators can be used to develop standalone damage functions at any geographic scale larger than 0.5 degrees or can be integrated directly into a larger integrated assessment model (IAM) framework. Emulators can also be used to study differences across crop models in responses to individual drivers of yield changes, making them useful for model comparison and improvement.

While an emulator that captures the response of a process-based crop model in a lightweight form will never be more *accurate* then its parent model, it can have multiple advantages over a numerical simulation. Emulation over the systematic sampling of the GGCMI Phase 2 experiment provides information on the influence of multiple interacting factors in a way that individual, more realistic process-based model runs cannot. Because we use a parametric statistical model, fitted parameter values can be physically interpreted to help understand differences between crop models. The flexibility and low computational requirements of emulators also make them particularly suitable for applications in integrated climate change impact assessments and projections of land-use change (e.g. Nelson et al., 2014a). Data storage requirements are reduced by three orders of magnitude: the yield output for a single crop model simulating all GGCMI Phase 2 scenarios for five crops is ~12.5 GB, while the equivalent global gridded emulator parameters are only ~20 MB. Computational requirements are nearly negligible: a thousand years of global 0.5 degree yields, i.e ~40,000,000 individual yield projections, can be emulated in 20 seconds on a laptop computer. The resulting suite of emulators should find considerable use in climate impacts analyses (e.g. Stevanović et al., 2016) and allow explicit evaluation of the uncertainty embedded in the choice of climate and crop models (Müller et al., 2017).

Several cautions should be noted when using the emulators presented here. First, extrapolation outside the GGCMI Phase 2 sample space should be avoided. Polynomial fits, while faithful within sample, quickly become non-physical outside of the tested range. This constraint is important given the strong warming expected under high-end greenhouse gas concentration scenarios (e.g. RCP8.5): if growing seasons are held fixed, climate model project mean temperature increases above 6K by end of century in many agricultural regions. Second, while the emulators are valuable for understanding the shape of yield responses and the factors that drive them, the absolute values of emulated yields should be treated with caution. The GGCMI Phase 2 models are not formally calibrated, so the emulators should be used for absolute projections only in combination with historical data. Third, neither growing season specification tested in GGCMI Phase 2 (A0 and A1) accounts for a major potential adaptation pathway under climate change, a shift to earlier or later planting dates (Waha et al., 2012), or generally

different growing seasons (Minoli et al., 2019a). And finally, the emulator should not be used to predict individual yearly yields, as the forced climatological mean yield response will not match the response to mean growing-season weather in a single year. The emulator cannot provide a measure of changing yield variance and should not be used to evaluate extremes.

In summary, the GGCMI Phase 2 dataset and emulators invite a broad range of potential future avenues of analysis. Future studies using the emulators described here could include a detailed examination of interaction terms, robust quantification
of model sensitivities to input drivers, and evaluation of geographic shifts in optimal growing regions. The large suite of crop models emulated lends itself particularly well to model comparison efforts, including identifying locations of model consensus (or lack thereof) and causes of model differences. While studies of yield responses to changes in growing-season variability would require new simulations, the emulators presented here provide a ready means of testing the null hypothesis that such effects are small. (Structured training sets could be constructed to directly study responses to variability changes:
see e.g. Poppick et al. (2016); Haugen et al. (2018) for methods of constructing synthetic climate timeseries with altered variability.) The GGCMI Phase 2 dataset can be used as a testbed for examining the ability of statistical models that use more detailed within-season regressors to capture both year-over-year and climatological changes, and for more systematic studies of emulation itself, including evaluation of alternate statistical specifications or machine learning methods. In general, the GGCMI Phase 2 experiment demonstrates the promise and utility of systematic parameter sweeps for improving understanding of the
factors driving crop responses and for evaluating and improving process-based crop models.

*Code and data availability.*   The polynomial parameters for crop model emulators are available at https://doi.org/10.5281/zenodo.3592453.

*Author contributions.*   J.E., C.M, A.R., J.F., and E.M. designed the research. C.M., J.J., P.F., C.F., L.F., R.C.I., I.J., C.J., W.L., S.O., M.P., T.P., A.Re., K.W., and F.Z. performed the simulations. J.F., J.J., A.S., M.L., Z.W., and E.M. performed the analysis and J.F., C.M., and E.M. prepared the manuscript. All authors contibuted to editing the manuscript.

*Competing interests.*   The authors declare no competing interests.

*Acknowledgements.*   We thank Michael Stein, Kevin Schwarzwald, and three anonymous reviewers, who provided helpful suggestions that contributed to this work. This research was performed as part of the Center for Robust Decision-making on Climate and Energy Policy (RDCEP) at the University of Chicago, and computing resources were provided by the University of Chicago Research Computing Center (RCC). This is paper number 36 of the Birmingham Institute of Forest Research.

*Financial support.* RDCEP is funded by NSF through the Decision Making Under Uncertainty program (grant #SES-1463644). James Franke was supported by the NSF NRT program (grant no. DGE-1735359) and the NSF Graduate Research Fellowship Program (grant #DGE-1746045). Christoph Müller was supported by the MACMIT project (grant no. 01LN1317A) funded through the German Federal Ministry of Education and Research (BMBF). Alex C. Ruane was supported by NASA NNX16AK38G (INCA) and the NASA Earth Sciences Directorate/GISS Climate Impacts Group. Christian Folberth was supported by the European Research Council Synergy (grant no. ERC-2013-SynG-610028) Imbalance-P. Pete Falloon and Karina Williams were supported by the Newton Fund through the Met Office program Climate Science for Service Partnership Brazil (CSSP Brazil). Karina Williams was supported by the IMPREX research project supported by the European Commission under the Horizon 2020 Framework program (grant no. 641811). Stefan Olin acknowledges support from the Swedish strong research areas BECC and MERGE, together with support from LUCCI (Lund University Centre for studies of Carbon Cycle and Climate Interactions). R. Cesar Izaurralde acknowledges support from the Texas Agrilife Research and Extension, Texas A & M University. Abigail Snyder was supported by the Office of Science of the U.S. Department of Energy as part of the Multi-sector Dynamics Research Program Area.

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
