# Peer review of "The GGCMI Phase 2 emulators: global gridded crop model responses to changes in CO2, temperature, water, and nitrogen (version 1.0)"

_Geoscientific Model Development, 2019_

## Referee Comment (RC1) · Anonymous Referee #1 · 31 Mar 2020

The study is a step forward in crop model emulation and is in principle a useful contribution to the literature. The paper contains a lot of excellent technical work. I focus here on areas for improvement, which I describe as major simply because I think a re-framing is needed in order to ensure that the paper is used well, and is not mis-used in the future.

The uses stated in the abstract for the emulators are: "providing a tool that can facilitate model comparison, diagnosis of interacting factors affecting yields, and integrated assessment of climate impacts." It would be good to understand more from the paper about how these different usages are envisaged. In particular, the suggestion that the

emulator might be used for integrated assessment lacks evidence. It is far from clear that this would be a sensible step to take, because study is subject to a number of important limitations. Whilst the authors are cognisant of these limitations, not enough attention is paid to them in the way that the work is framed and interpreted.

One limitation is the use of mean yields. "We emulate the climatological mean response, because that is the response of interest in assessments of climate change impacts. . .. Emulation then becomes relatively straightforward, since changes in time-averaged yields are also considerably smoother than those in year-to-year yield response." – L108. Why is mean yield the response of interest? Perhaps it is because it is relatively straightforward, rather than because it is useful per se. Climate variation explains a third of global crop yield variability – Ray et al. (2015) Nature communications. Why do the authors think that mean yields are interesting? There would need to be a clear rationale in the paper.

Assuming a rationale exists for assessing mean yields, over what lead times might the emulators be usefully used? As the authors point out, the emulators cannot be used out of sample, thus implying relatively short lead times, before climate changes significantly. However, over the next couple of decades, changes in mean yields are unlikely to be important relative to extremes.

Assuming a focus on mean yields can be justified for an appropriate lead time, there remains the question of why an emulator is a valid method to use. Two issues need to be addressed here:

i. Whether or not the emulator is fit for purpose. Does it reproduce observed yields well? The link to observed yields is tenuous. Error (which should actually be termed "deviation" – since it is not a true error) is defined relative to yields simulated by the underlying crop models. If the emulators are to be used, then one would need to be sure it captures real historical climate impacts. The language on this is imprecise in many places. For example, in the abstract: "... suggesting that effects of changes

in temperature and precipitation distributions are small relative to those of changing means." This statement is true only of model space; indeed, it is untrue of observations as Ray et al. (2015) and others have pointed out.

ii. Is there a better method? Statistical regressions would by definition capture to some extent observed yield responses to weather and climate. The resulting emulators [are] lightweight, computationally tractable " – but so are statistical models. Reasons to use an emulator over a statistical model are presented in the introduction. However, neither the lack of observed yields in calculating skill, nor the lack of model calibration (another limitation; see below), are brought into this discussion. Similarly, what does the focus on yield changes, rather than yields per se, mean for the robustness of the methodology?

The other option discussed briefly in the introduce is the use of process based models. The full set of GGCMI simulations is available; surely the emulators are not expected to outperform their masters? Presumably the "lightweight" approach is deemed to be an advantage for integrated assessment. If this is so then the advantage should be clearly presented.

The major revisions needed for the paper will follow on naturally from framing it more clearly to demonstrate the uses the emulators can be put to. As is no doubt clear, I think that the rationale for their use in integrated assessment is extremely difficult to demonstrate; but perhaps I am wrong. It would be worth thinking about the conditions (data availability, crop knowledge, model skill, input data availability, ..) under which the emulators might be a preferred option.

Model comparison and diagnosis are easier to justify – but even here some work is needed to explain how the emulators could be used. The emulators could be used to highlight areas of CTWN-A where there is consensus and where there is not, thus providing clear evidence of where model improvement, and associated observational datasets, are needed.

---

## Referee Comment (RC2) · Anonymous Referee #1 · 31 Mar 2020

Methodology is not clearly separated from results More information on the skill of the models that go into emulators would aid rationale. Some models are more skilful than others. Do you expect the MME to be the most skilful simulation? If different models perform better in different regions, why not use this information in the emulators?

Similarly, which processes are included vs not included in the underlying models. How good at threshold responses are these models? Cf "In general, emulator performance is poor anywhere that models show steep yield changes once some threshold has been reached, whether these are abrupt gains or complete crop failures" - I find these cases very important especially when looking at the end of the century.

Why different numbers of perturbations used across different models?

Use of normalised "error" (should be "deviation" or similar) makes differences between models hard to see and makes results appear perhaps better than they are.

Be clear which data were used for calibration vs evaluation

"Emulator performance is generally good relative to model spread in areas where crops are currently cultivated and in temperate zones in general" - probably not hard giving that the crop models are not calibrated. I think the whole study should have been done with calibrated crop models.

line 115 - put info for figure in caption as not helpful in main text. And in Fig.1 , cannot see advertised labels of a, b, c, d (although perhaps journal adds these later).

line 195 onwards - would model features that are able to be dropped be the same if the procedure was repeated including non-cultivated land? i.e. in marginal areas, are different factors important for determining yields? This is mentioned below (line 223).

---

## Referee Comment (RC3) · Anonymous Referee #2 · 1 Apr 2020

Overview:

Understanding crop yield response to environmental changes is crucial for food security. Statistical crop models are easier for calculation but the projection capability is constrained by the range of current conditions. The process based crop models aim to capture the yield response to different environmental changes but computational expensive compared to statistical crop models. This study developed statistical emulators for 9 process based crop models using GGCMI phase II simulations. The author well validated the statistical emulators and discussed the caveats and the potential usage, such as provide an alternate approach for impact assessment. The manuscript is generally well written and I only have several minor comments on the method and results.

Minor comments:

The whole section 2.2 discussed why there are differences in climatological and year-to-year response. This part is very interesting but somehow could divert the readers who are eager to know how the study uses the training data to develop emulators. It could be a better flow if put the section into the discussion section or supplementary.

The authors need to refine the section 3.1 to give more information on Y and regressors (what temporal and spatial scale). Line 161 mentioned that "Emulating at the grid cell level". So I think equation 1 was fitting at grid cell level. My understanding is that Y is a vector of 30-year averaged crop yields across different uniform changes scenarios (a total of 756 scenarios?) in one GGCMI model. There are 34 terms in equation 1, aren't there over fitting problems when you have a small number of Y (some models did not done all required scenarios) but a large number of regressors. Please comment.

Line 138: "in the the". Double the here.

Equation 2. Some terms are gray in equation 2. Are those the dropped terms? If so, just delete them.

Figure 10 caption. "the five GCCMI Phase II crops", the authors used this terms several times, but this sounds like there are five special crops that was created by GGCMI Phase II. I think just say five crops is fine. They are common crops. And how many individual models are incorporated here? I guess it is nine. But there are not nine color lines, is that because some lines are underneath the black thick line? If so, please mention that.

Figure 11. In the figure legend, the uniform T sounds like each process model was forced with global uniform T. But I think it means the uniform increase of T, uniform $\Delta T$ is better.

Figure 11. In the caption, "Circles are emulated yearly global production changes", those are dots, not circles.

Figure 11. Why there are no open squares on plot b? And in plot c, open squares for 2 and 4 increasing of T is missing. All the three plots showed emulated uniform T lines, why not show emulated uniform T+W for plot b, and emulated uniform T+W+C for plot c?

In the SI:

Page 2. First line "is not uniform tn the GGCMI Phase II", what is tn? Should be in?

Figure S6: there are no gray lines (Ontario), why? I want to know if Ontario has the same failure in A1 as in A0.

Figure S21: The simulated RCP8.5 (open triangle ) were not found on the graph.

---

## Referee Comment (RC4) · Anonymous Referee #3 · 13 Apr 2020

The authors present a highly detailed description and evaluation of newly-developed statistical emulators for global gridded crop model simulations (as being contributed to the GGCMI Phase II), specifically targeting emulation of mean yield changes due to changed climate conditions. The authors construct these emulators by varying over carbon dioxide concentrations, temperature, water, and nitrogen inputs, and also test the effects of adaptation.

In general, this paper is highly useful contribution to the emerging work of global gridded crop modeling primarily due to providing a very well tested, relatively low-error, computationally economical, and low data-input means of reproducing and/or running

GGCM experiments (again, as related to GGCMI Phase II). Given the computational expense and large data requirements of the GGCMs, it is worthwhile to have an option to run climate-crop experiments with comparatively less "overhead" and relatively high confidence that the emulators overall faithfully represent specific model and (thus ensemble?) sensitivities. I also think that the authors generally did well to note some key uncertainties both in the GGCMs and how these influence the emulators, although a couple of aspects could be addressed a bit more (and I note these below).

Ultimately, one of the key strengths of this work is to provide a comparable and easier means of representing geospatial crop responses relative to the GGCMs (which certainly have other uses as full process-models). Thus, with a few minor revisions, this paper makes an interesting and useful contribution to the field, and I anticipate these emulators being put to good use by many researchers exploring mean climate-crop interactions.

I do have just a few questions and remarks that may be useful to the authors as they think about some minor revisions and next steps:

Section 3.2, Lines 215 onward: I found it interesting that several of the carbon-terms dropped out due to their relatively negligible contributions. [$CO_2$] effects on crops (and ecosystems!), and their nonlinear interactions with other changing climate parameters, are still highly uncertain. Crop models also display much variability in their respective [$CO_2$] responses. I noticed that for the simulations emulating HadGEM responses [$CO_2$] was held fixed or not varying with other parameters. Since the authors are emulating, and evaluating against, GGCM outputs, if the GGCMs do not display [$CO_2$], then it follows that neither will the emulators I suppose. However, I wonder if the authors could further comment on this: the fact that [$CO_2$] was negligible for the emulation does not necessarily mean the effects are negligible in reality, correct? I don't see this discussed much elsewhere in the manuscript, so having a bit more commentary on this, with respect to [$CO_2$] and/or more generally, would be useful as readers consider the terms of your model emulator.

Section 4 and elsewhere: This comment is not just relegated to Section 4, but I'm more generally trying to parse out the relative contributions of climate variability and mean climate change, and the arguments provided in the paper that support emulation of the latter. I think there are two types of "variability" (admittedly not the best word, perhaps more "characteristics" other than the climatological mean yield) that the authors address that might be clarified just a bit more in the Discussion to avoid any confusion.

Firstly, in Section 2.2, the authors make the case that year-to-year variability is structurally different than simulation of the climatological mean yields, and that the former doesn't preclude the latter, correct? The authors also highlight (I think) that the emulators are not suitable for a full interpretation of interannual variability and extremes, particularly highly non-linear interactions between climate (and other) parameters, despite the higher order terms of the emulation (which, as the authors note, are geared towards emulating climatological means).

Secondly, in Section 4.3, the authors demonstrate that potential shifts in the distributions of climate parameters do not impact the climatological yield emulations and the results still compare well with the GGCMs (Figure 9). This fact – this shift in distribution and potential changes in variability that result from it (which is where the readers' mind might go, as mine certainly did) – is I believe distinct from the discussion of year-to-year variability discussed above.

I appreciate that the authors have provided detailed explanations of their approach, treatment, and findings wrt to considering these variability and distributional changes. Still, there's a lot of material here to keep track of, and I think it may be useful to reiterate each of the above points clearly in the Discussion (particularly if I've mistakenly represented it, as I think this may be an example of reader confusion!). For example, there is a sentence in the Discussion that minimizes the impact of future variability (particularly at the aggregate level – around Line 445), particularly in the area aggregate, and I think this is in reference to the findings in Section 4.3 However, this doesn't mean that interannual variability, or extremes or nonlinear interactions, won't be impactful to

future (or current) crop impacts.

Discussion: Lastly, I think the major point of this paper is to provide these emulator frameworks as an alternative to climate-crop assessments with the full process-based GGCMs. I therefore understand the authors' approach to evaluate the emulators against the GGCMs – this is quite reasonable.

It might be helpful, though, to take one step beyond this and compare to some observed historical yield changes. I would not expect this to be better than the GGCMs, and such evaluations have already been done for the GGCMs, so I would expect to see a similar response (and this is notwithstanding the applicability and veracity of comparison products). However, I don't think I've seen such an evaluation for GGCMI Phase 2 yet (I expect one is planned), and so pre-empting this with a comparison of the emulators may just be useful to have on hand. If this could be done and stuck into Supplementary, it would be a useful figure for the community moving forward, rather than having to show an intermediary figure of emulator-GGCM comparisons.

---

## Author Response (AR1)

Author response to comments on gmd-2019-365

Referee comments in blue | Author response in black | MS change in red (line numbers refer to those in the attached latex diff document)

**Referee 1:**

 The study is a step forward in crop model emulation and is in principle a useful contribution to the literature. The paper contains a lot of excellent technical work. I focus here on areas for improvement, which I describe as major simply because I think a re-framing is needed in order to ensure that the paper is used well, and is not mis-used in the future.

Thank you for the assessment. We have added text in accordance with suggestions below.

2. The uses stated in the abstract for the emulators are: "providing a tool that can facilitate model comparison, diagnosis of interacting factors affecting yields, and integrated assessment of climate impacts." It would be good to understand more from the paper about how these different usages are envisaged. In particular, the suggestion that the emulator might be used for integrated assessment lacks evidence. It is far from clear that this would be a sensible step to take, because study is subject to a number of important limitations. Whilst the authors are cognisant of these limitations, not enough attention is paid to them in the way that the work is framed and interpreted.

We have added text in line with these suggestions; see responses below.

3. One limitation is the use of mean yields. "We emulate the climatological mean response, because that is the response of interest in assessments of climate change impacts. . .. Emulation then becomes relatively straightforward, since changes in time- averaged yields are also considerably smoother than those in year-to-year yield response." – L108. Why is mean yield the response of interest? Perhaps it is because it is relatively straightforward, rather than because it is useful per se. Climate variation explains a third of global crop yield variability – Ray et al. (2015) Nature communications. Why do the authors think that mean yields are interesting? There would need to be a clear rationale in the paper.

We believe that changes in multi-annual averages are actually the most useful measure of future interest. While changes to year-to-year variability in crop yields would be important to farmers if they change significantly, the shift that is most relevant to overall economic impacts, and to decisions on choice of crops and planting locations, is that in mean yields. Many climate change impact assessments therefore focus on multi-annual means as the central metric for climate change impacts. In economic assessments that use crop model outputs to inform IAMs or agro-economic land-use models, crop model outputs are also typically aggregated to multi-annual means (Nelson et al. 2014, Wiebe et al. 2015), because land-use changes (in terms of expanding or abandoning cropland) are driven not by short-term (year-to-year) yield variability, but by changes in average conditions. Finally, it is clear that year-to-year variability in yields is only loosely related to mean growing season temperatures, which are the dominant changes in the underlying dataset. In process-based crop models, changes in mean yields are tightly

related to mean temperatures, so we can provide a reliable emulator of these changes. We therefore have focused our efforts on prediction of changes in mean yields.

The reviewer's comment shows that we have not adequately discussed these considerations, and so we have added text along these lines. We have also clarified that the application of our emulators is constrained to research questions in which long-term dynamics are the relevant feature and that short-term dynamics need to work with other tools. We have also added in our concluding sections more discussion of what would be needed to analyze and emulate changes in crop yield variability, for those working with a focus on these time-scales (e.g. Schewe et al. 2017).

Additional text pertaining to the choice of mean yields has been added to Section 2.2, Lines 118 - 125.

Additional text addressing the issue of Ray et al added to line 400.

4. Assuming a rationale exists for assessing mean yields, over what lead times might the emulators be usefully used? As the authors point out, the emulators cannot be used out of sample, thus implying relatively short lead times, before climate changes significantly. However, over the next couple of decades, changes in mean yields are unlikely to be important relative to extremes.

We agree that year-to-year variability is likely more interesting over the next decade (or two). The emulator is designed to provide projections at the decadal or multidecadal timescale to the end of the century (following in line with the RCP-IPCC framework). While it is true that some areas of the globe will exceed 6 degrees at the high end of climate change (e.g. RCP8.5) by the end of the century, this is not the case for many regions or scenarios with lower radiative forcing, especially when considering changes in multi-annual averages. The projection to the end of the 21st century with assumed fixed management, such as the growing season is unrealistic anyway and needs to be interpreted with care (Minoli et al. 2019, lizumi et al. 2019).

We have added additional language to the discussion to clarify this timescale of interest, the problems with extrapolation, and the limitations with the fixed growing season.

Additional text pertaining to the choice of mean yields has been added to Section 2.2, Lines 118 - 125.

Additional cautions about extrapolation have been added to Section 6, lines 544-547.

Additional notes about the fixed growing season have been added to Section 6, lines 551-553.

- 5. Assuming a focus on mean yields can be justified for an appropriate lead time, there remains the question of why an emulator is a valid method to use. Two issues need to be addressed here:
- 6. i. Whether or not the emulator is fit for purpose. Does it reproduce observed yields well? The link to observed yields is tenuous. Error (which should actually be termed "deviation" since it is not a true error) is defined relative to yields simulated by the underlying crop models. If the emulators are to be used, then one would need to be sure it captures real historical climate impacts. The language on this is imprecise in many places. For example, in the abstract: "... suggesting that effects of changes in temperature and precipitation distributions are small relative to those of changing means." This statement is true only of model space; indeed, it is untrue of observations as Ray et al. (2015) and others have pointed out.

The purpose of the emulators is to reproduce the output of the process-based models, to provide a lightweight substitute for the computationally expensive calculations. We therefore do not focus on validation of those process models in this paper. Extensive model validation exercises were carried out as part of GGCMI Phase I (Müller et al. 2017), and we address model validation of GGCMI Phase II in the "experiment description" GMD paper (Franke et al 2020a). The emulator is therefore fit for purpose if it captures the output of the process-based models, which we cover here extensively in Section 4.1.

We agree absolutely that variability in growing-season conditions is critical for year-over-year yield variations. This was shown with historical yields in the Ray 2015 study, and is also true for the process-based models here (see Franke et al 2020a). However, we are unaware of any studies showing that **changes** in variability under climate change are important compared to changes in climate means. Capturing the effects of changing variability in climate projections would be problematic in any case because climate models show very little agreement about future changes in variability (compared to their agreement in the change in means), and often struggle to represent historical variability.

Our statement about the ability of our emulators to capture mean yields in a process-based model under a climate model projection, inclusive of any variability changes, is a demonstrably true statement for the GGCMI simulations. It remains an interesting question whether the process-based models are less sensitive to potential future changes in temperature and precipitation distributions than are real-world crops. Some suggestions along these lines were made by Müller et al 2017, which is cited in the manuscript, but we have now clarified the finding.

In general, these comments suggest that we have inadequately discussed the underlying differences between the process-based models used in GGCMI and statistical models based on historical crop yields. We have therefore now better emphasized these points in the manuscript.

**Additional notes about the application of the emulator have been added to Section 6, lines 495-530.**

7. ii. Is there a better method? Statistical regressions would by definition capture to some extent observed yield responses to weather and climate. The resulting emulators [are] lightweight, computationally tractable " – but so are statistical models. Reasons to use an emulator over a statistical model are presented in the introduction. However, neither the lack of observed yields in calculating skill, nor the lack of model calibration (another limitation; see below), are brought into this discussion. Similarly, what does the focus on yield changes, rather than yields per se, mean for the robustness of the methodology?

Statistical models are being developed by many different research groups and consist of a separate and somewhat distinct approach to process-based modeling. Statistical models have the obvious problem of little data in many geographic regions and no data under future climate change that has not yet happened. Statistical models can only be evaluated on 'held-out' historical data. It is unclear (and perhaps impossible to test) whether statistical models or process-based models are better for future projections.

The emulator is a statistical model, only it is trained on simulated data instead of 'real' data. It has the obvious advantage of leveraging the body of science behind crop models to provide 'data' where none exist in the real world, both in space (where crops are not currently grown) and time (under climate change that hasn't happened yet). Better forms of emulation may be possible within the GGCMI phase II

framework. We hope the simulation output dataset can become a test-bed for investigating different statistical functional forms.

While we do fit an intercept (historical mean yield), the emulator is intended to be coupled with a dataset of actual yields since models are uncalibrated. We therefore stand by the focus on yield change as a better use of the emulator for impact assessment. We feel this is a more robust application of the emulator.

We have added some additional text to the text to discuss these issues.

**Additional notes about calibration added to lines 111-113.**

8. The other option discussed briefly in the introduce is the use of process based models. The full set of GGCMI simulations is available; surely the emulators are not expected to outperform their masters? Presumably the "lightweight" approach is deemed to be an advantage for integrated assessment. If this is so then the advantage should be clearly presented.

Correct, the emulator cannot be better than the model it is trained on. The advantage of the emulators is that by providing an analytical form for yield based on climate and nutrients, they allow simulating yields quickly under arbitrary climate forcing scenarios, as would be needed in a study of optimal policies addressing climate change, or in an assessment exercise using non-standard climate projections. Even large sets of pre-computed crop model outputs lack the flexibility to be adjusted to the applications' needs.

The GGCMI Phase II simulations would not typically be used in assessments directly, since they consist of non-physical combination of parameters and non-physical spatial distributions in climate changes. No single simulation represents a plausible future world, but in combination they allow production of an emulator that can capture yield response under many plausible future scenarios.

We have added some additional discussion on this topic.

**Additional notes have been added to Section 6, line 528 - 531.**

9. The major revisions needed for the paper will follow on naturally from framing it more clearly to demonstrate the uses the emulators can be put to. As is no doubt clear, I think that the rationale for their use in integrated assessment is extremely difficult to demonstrate; but perhaps I am wrong. It would be worth thinking about the conditions (data availability, crop knowledge, model skill, input data availability, ...) under which the emulators might be a preferred option.

We have added text discussing their use in Integrated Assessment Models (IAMs). As described above, an emulation of climatological mean yield is the appropriate input for IAMs and other economics-based land-use models whose land-use dynamics are always, to our knowledge, based on multi-annual mean yields. It would in fact be incoherent for an IAM to make decisions about land-use changes based on yearly yields, since most IAMs we are aware of utilize climatological mean temperature changes as their climate inputs (sometimes only the global average). As mentioned above, mean temperature change is closely related to mean yield change but only very loosely related to yearly yield variations.

The emulators presented here are developed in collaboration with IAM modelers to meet their needs; please note that the co-author list includes IAM modelers. We recognize though that the submitted

manuscript did not sufficiently emphasize the expected end uses, and so have now worked with our co-authors to add new text describing the several projects currently in development integrating these emulators into IAMs.

**Text added to section 2.2, line 119 and section 6, line 533 - 535 to address IAM integration.**

10. Model comparison and diagnosis are easier to justify – but even here some work is needed to explain how the emulators could be used. The emulators could be used to highlight areas of CTWN-A where there is consensus and where there is not, thus providing clear evidence of where model improvement, and associated observational datasets, are needed.

Indeed, model comparison and diagnosis is one of the primary intended applications. Several publications are currently in preparation that use the emulators described here for just these purposes: studies that diagnose differences in model responses to particular climate and management inputs, or clarify the interactions between parameters (e.g temperature and precipitation, or temperature and nitrogen addition). These studies are not possible using statistical models fit on historical yields, but require process models run over systematic parameter sweeps. We had discussed this in the Introduction, but as the paper is long we realize that it requires additional text in the Conclusions/Discussion describing these studies, and have added this.

Text added to Section 6, lines 526, 531, 560.

11. Methodology is not clearly separated from results More information on the skill of the models that go into emulators would aid rationale. Some models are more skilful than others. Do you expect the MME to be the most skilful simulation? If different models perform better in different regions, why not use this information in the emulators?

The skill of the underlying crop models is described and discussed in the companion paper (Franke et al. 2020a) and in earlier efforts to describe the crop models' skill (Müller et al. 2017). The paper under review is intended as the "model description" paper describing the development of emulators, not a documentation of the process models themselves. The question of if and under which conditions the MME is the most skilful simulation is a question about the process models themselves. This paper focuses on validating the emulators, i.e. on showing that a simple functional form can capture the response of those process models. The emulators can then become a tool for answering exactly the question that the reviewer poses, and we appreciate the suggestion. Note that the emulators are designed for each crop model individually and can be combined and aggregated at the users' choice and needs for specific applications. We have now added text suggesting that emulators can be used to examine regional model performance.

**Text added to Section 6, lines 560.**

12. Similarly, which processes are included vs not included in the underlying models. How good at threshold responses are these models? Cf "In general, emulator performance is poor anywhere that models show steep yield changes once some threshold has been reached, whether these are abrupt gains or complete crop failures" - I find these cases very important especially when looking at the end of the century.

Indeed. These cases are important and the provision and publication of the emulators, that are described here, allows for making these analyses. Again, this paper is a model description paper, not the final

application of the emulators that could answer all questions that could be addressed by using the emulators. As discussed above and in the paper, one intended purpose of the emulators is to scrutinize model dynamics and identify options for model improvement (of the process-based crop models, not the emulators).

The temperature response at the 30-year mean scale is very smooth in all but a few cases. Discontinuities (steep changes) in yield are more common when some models show no yield under present conditions and then transition to moderate yield under a certain amount of warming. While some thresholds may exist on the high end for temperature at the yearly scale, there are vanishingly few cases where the 30-year mean yield drops to zero under warming.

We have added some text pointing out some of these cases to clarify the point.

Text added to section 4.2, line 346-347.

**13. Why different numbers of perturbations used across different models?**

The complete set of simulations is computationally very demanding, and so modeling groups were offered a set of participation "tiers" involving different number of simulations. The protocol is described in detail in the companion "experiment description" paper (Franke et al. 2020a). We have added text here to point the reader to this documentation.

Text added to section 2.1, line 112.

**14. Use of normalised "error" (should be "deviation" or similar) makes differences between models hard to see and makes results appear perhaps better than they are.**

We agree that the normalised error does not provide complete information, but it is a useful metric in the context of multi-model emulation, because it normalizes the errors in those regions where models disagree quite strongly anyway. Put another way, this metric emphasizes the need for faithful emulation of model output in those places where the models best agree.

Note that we have included in the paper a separate metric that is not normalized across models: the "out-of-sample evaluation". This test treats all models equally and as separate entities, and was included specifically to provide the kind of assessment the reviewer seeks.

We have added language better clarifying the differences between the two separate evaluation methods and now clarify the difference between 'errors' and 'deviations'.

**Text added to section 4, line 274.**

**15. Be clear which data were used for calibration vs evaluation**

We assume that 'calibration' here refers to the emulator out-of-sample evaluation process. For out-of-sample validation, we use a 3-fold cross validation procedure where 90% of the data are used to train the emulator (calibration) and the held out 10% is used to evaluate. This is repeated two more times and the results are averaged. The exact simulation cases in each fold vary by model depending on which were provided, and would be quite exhaustive to list in detail. For example for a single model, this would

consist of three lists of 675 conditions that were included in training and 75 conditions that were evaluated against. We do not think such a table of 2200 different listed conditions would be very illustrative.

We have updated the text to make the cross-validation procedure more clear.

Text added to clarify the procedure section 4.2, lines 373-379. There was an error in our original response letter. The cross validation is 90% - 10%, not 2/3 - 1/3 as originally stated. The manuscript was correct.

16. "Emulator performance is generally good relative to model spread in areas where crops are currently cultivated and in temperate zones in general" - probably not hard giving that the crop models are not calibrated. I think the whole study should have been done with calibrated crop models.

As with so many things, there are pros and cons with calibration, especially if no suitable calibration target is available. Calibration would be needed if the intent of the exercise were to produce absolute yields. However, we are focused here on understanding model responses to different climate and management inputs, and in forecasts of fractional changes. We feel that those are adequately and perhaps better addressed with uncalibrated models. In the previous Phase of GGCMI (Elliott et al. 2015, Müller et al. 2017), the harmonization of management conditions appeared to lead to very different model behavior in some models. Note also that global-scale crop model calibration poses tremendous challenges given the lack of calibration targets (see e.g. Müller et al. 2017).

The lack of calibration may make our 'normalised error' metric less stringent than it might be, since calibration would likely (but not necessarily) reduce the spread between models. (See e.g. Müller et al. 2017 for discussion of the effects on future projections of calibration to present-day yields.) But, the normalised error is only one means of assessing emulators, and we conduct the second, non-normalised 'out of sample' validation exercise to provide an assessment independent of the inter-model spread.

We have expanded the text on the rationale for using uncalibrated models, and implications for the application and interpretation of the emulators.

**Text added to section 2, lines 111- 113.**

17. line 115 - put info for figure in caption as not helpful in main text. And in Fig.1 , cannot see advertised labels of a, b, c, d (although perhaps journal adds these later).

Modified as suggested.

Text removed, as suggested, from lines 129-131.

18. line 195 onwards - would model features that are able to be dropped be the same if the procedure was repeated including non-cultivated land? i.e. in marginal areas, are different factors important for determining yields? This is mentioned below (line 223).

The suggestion of doing feature selection over non-cultivated land was not tested in this study, but it is an interesting question and could be pursued in follow-up studies, since we provide the full and the reduced form of the emulators. We hope that others will extend on the work shown here.

We have added this point to the Discussion.

Text added to line 560.

**Referee 2:**

**Overview:**

 Understanding crop yield response to environmental changes is crucial for food security. Statistical crop models are easier for calculation but the projection capability is constrained by the range of current conditions. The process based crop models aim to capture the yield response to different environmental changes but computational expensive compared to statistical crop models. This study developed statistical emulators for 9 process based crop models using GGCMI phase II simulations. The author well validated the statistical emulators and discussed the caveats and the potential usage, such as provide an alternate approach for impact assessment. The manuscript is generally well written and I only have several minor comments on the method and results.

**Thank you for the assessment.**

**Minor comments:**

2. The whole section 2.2 discussed why there are differences in climatological and year-to-year response. This part is very interesting but somehow could divert the readers who are eager to know how the study uses the training data to develop emulators. It could be a better flow if put the section into the discussion section or supplementary.

We know the paper is very long, but we felt this section was necessary here to explain the rationale for developing our emulators at the climatological mean level. This is a key feature of the study and is a point of confusion for other reviewers and readers. We hoped that by separating this discussion into its own sub-section, readers would feel free to skip it if they do not feel that the choice of the climatological mean yield requires justification. We have tried to add a little more structure to the introduction to allow readers to better pick and choose which sections to focus on, and following another reviewer's suggestion, have now tried to better recap the main points of the paper in the Discussion.

**Text added to section 1, lines 83 - 86.**

**Recap added to section 6, lines 496 - 527.**

3. The authors need to refine the section 3.1 to give more information on Y and regressors (what temporal and spatial scale). Line 161 mentioned that "Emulating at the grid cell level". So I think equation 1 was fitting at grid cell level. My understanding is that Y is a vector of 30-year averaged crop yields across different uniform changes scenarios (a total of 756 scenarios?) in one GGCMI model. There are 34 terms in equation 1, aren't there over fitting problems when you have a small number of Y (some models did not done all required scenarios) but a large number of regressors. Please comment.

Indeed, the equation was fitted at grid cell level. Overfitting can be a problem, and some models could not be emulated if they provided too few simulations to the GGCMI Phase 2 simulation data set. We felt that the number of simulations provided was sufficiently important that we repeat Table 3 from the companion paper Franke et al. 2020a that describes the GGCMI Phase 2 experimental protocol.

In the best case, the training domain consists of 756 elements in Y, which is more than sufficient for fitting 34 parameters, according to a "one in ten rule". Not all models have provided the full sample, but we use a Bayesian regularization scheme (that probabilistically weights parameters towards zero) that mitigates overfitting in the cases with fewer samples. The out-of-sample validation is our test of whether overfitting is a problem - we show that the emulators fit with the Bayesian scheme can predict yields not included in the training set even in the model cases with lower sampling, but that overfitting would be a problem with standard OLS. We have expanded the text in this section to better explain the overfitting concerns and why we think they are addressed.

**Text added to section 3.1, lines 185 - 189.**

4. Line 138: "in the the". Double the here.

Thanks for the catch. Removed.

5. Equation 2. Some terms are gray in equation 2. Are those the dropped terms? If so, just delete them.

Terms in gray here are dropped. We left them for clarity of comparison. We have added some language to make this more clear.

**Text added to line 215.**

6. Figure 10 caption. "the five GCCMI Phase II crops", the authors used this terms several times, but this sounds like there are five special crops that was created by GGCMI Phase II. I think just say five crops is fine. They are common crops. And how many individual models are incorporated here? I guess it is nine. But there are not nine color lines, is that because some lines are underneath the black thick line? If so, please mention that.

We will modify the language to remove the GGCMI designation as suggested.

All models are included in this figure, but not all models provided simulations for all crops and not all models provided simulations across the nitrogen dimension, so the number of lines is less than 9 in some cases. We now state this explicitly in the figure caption.

**Text modified in the Figure 10 caption.**

7. Figure 11. In the figure legend, the uniform T sounds like each process model was forced with global uniform T. But I think it means the uniform increase of T, uniform DT is better.

Agreed, this is an excellent point. Modified as suggested.

'Delta' added to figure 11 caption as suggested.

8. Figure 11. In the caption, "Circles are emulated yearly global production changes", those are dots, not circles.

Agreed. Modified as suggested.

"Circles" changed to "dots" in figure 11 caption and where referenced in the text.

9. Figure 11. Why there are no open squares on plot b? And in plot c, open squares for 2 and 4 increasing of T is missing. All the three plots showed emulated uniform T lines, why not show emulated uniform T+W for plot b, and emulated uniform T+W+C for plot c?

To clarify: the open squares are not emulations, they are the actual simulation output. The emulated responses are the solid dots.

Note that this figure does not involve process model simulations of yield under future climate projections. Instead, it shows *emulations* of yields under climate projections, and compares these emulated yields to the uniform-offset simulations of the GGCMI phase II dataset.

In the case where only temperature is allowed to change, we can show a simulation that is a direct analogue for an emulation of a climate projection. In the T and W case, both temperature and precipitation are changing in the climate projection, and we have no equivalent uniform-offset crop simulations. We cannot match the simultaneous values of T and W changes.)

We recognize that this figure is complex and the caption is not as clear as it could be. We have adjusted the language to try to better explain what is being shown.

Text added to Figure 11 caption.

In the SI:

10. Page 2. First line "is not uniform th the GGCMI Phase II", what is the Should be in?

Should be 'in'. Corrected.

Correction made in supplement.

11. Figure S6: there are no gray lines (Ontario), why? I want to know if Ontario has the same failure in A1 as in A0.

Good suggestion; the requested line has been added to the figure.

PROMET (ontario) line added to Figure S6 in the supplement.

12. Figure S21: The simulated RCP8.5 (open triangle ) were not found on the graph.

This was in error. Caption modified.

Legend modified in Figure S21 in the supplement.

**Referee 3:**

1. The authors present a highly detailed description and evaluation of newly-developed statistical emulators for global gridded crop model simulations (as being contributed to the GGCMI Phase II), specifically targeting emulation of mean yield changes due to changed climate conditions. The authors construct these emulators by varying over carbon dioxide concentrations, temperature, water, and nitrogen inputs, and also test the effects of adaptation. In general, this paper is highly useful contribution to the emerging work of global grid- ded crop modeling primarily due to providing a very well tested, relatively low-error, computationally economical, and low data-input means of reproducing and/or running GGCM experiments (again, as related to GGCMI Phase II). Given the computational expense and large data requirements of the GGCMs, it is worthwhile to have an option to run climate-crop experiments with comparatively less "overhead" and relatively high confidence that the emulators overall faithfully represent specific model and (thus ensemble?) sensitivities. I also think that the authors generally did well to note some key uncertainties both in the GGCMs and how these influence the emulators, although a couple of aspects could be addressed a bit more (and I note these below). Ultimately, one of the key strengths of this work is to provide a comparable and easier means of representing geospatial crop responses relative to the GGCMs (which certainly have other uses as full process-models). Thus, with a few minor revisions, this paper makes an interesting and useful contribution to the field, and I anticipate these emulators being put to good use by many researchers exploring mean climate-crop interactions.

Thank you for the assessment.

I do have just a few questions and remarks that may be useful to the authors as they think about some minor revisions and next steps:

2. Section 3.2, Lines 215 onward: I found it interesting that several of the carbon-terms dropped out due to their relatively negligible contributions. [CO2] effects on crops (and ecosystems!), and their nonlinear interactions with other changing climate parameters, are still highly uncertain. Crop models also display much variability in their respective [CO2] responses. I noticed that for the simulations emulating HadGEM responses [CO2] was held fixed or not varying with other parameters. Since the authors are emulating, and evaluating against, GGCM outputs, if the GGCMs do not display [CO2], then it follows that neither will the emulators I suppose. However, I wonder if the authors could further comment on this: the fact that [CO2] was negligible for the emulation does not necessarily mean the effects are negligible in reality, correct? I don't see this discussed much elsewhere in the manuscript, so having a bit more commentary on this, with respect to [CO2] and/or more generally, would be useful as readers consider the terms of your model emulator.

We agree that it is highly interesting that the higher-order interaction terms could be dropped for most models, and hope that this will be the subject of a follow-up paper. This type of finding is part of what makes emulators powerful as a diagnostic tool of model behavior. Two models (PROMET and JULES) required the higher order CO2 interactions for accurate emulation, and it would be interesting to understand why.

Note that the magnitude of the pure CO2 terms is very large. The CO2 response is critical and results in large yield changes. (See Figure 10 for example).

In the HadGEM simulations shown in Figure 9, we held out CO2 precisely because the crop CO2 response is large and the purpose of this exercise was to examine the fidelity of the emulators' temperature / precipitation response. Figure 9 examines whether an emulator trained on the GGCMI Phase II database, which allows for no changes in climate variability, can accurately reproduce crop yields under actual climate model output, that may involve some changes in variability as well as means. We agree that this issue is under-discussed and have now added text to explain this more carefully. We now explicitly note that the CO2 response in LPJmL is so large that it almost completely negates the damages caused by higher temperatures, and that we hold it out to isolate the temperature-driven response.

**Text added to section 4.3, line 396 - 415.**

3. Section 4 and elsewhere: This comment is not just relegated to Section 4, but I'm more generally trying to parse out the relative contributions of climate variability and mean climate change, and the arguments provided in the paper that support emulation of the latter. I think there are two types of "variability" (admittedly not the best word, perhaps more "characteristics" other than the climatological mean yield) that the authors address that might be clarified just a bit more in the Discussion to avoid any confusion. Firstly, in Section 2.2, the authors make the case that year-to-year variability is structurally different than simulation of the climatological mean yields, and that the former doesn't preclude the latter, correct? The authors also highlight (I think) that the emulators are not suitable for a full interpretation of interannual variability and extremes, particularly highly non-linear interactions between climate (and other) parameters, despite the higher order terms of the emulation (which, as the authors note, are geared towards emulating climatological means).

Correct. Climatological mean yields are closely related to climatological mean temperature. Year-to-year yields are driven by weather factors other than (or in addition to) mean temperature. We show in Figure1 that regressing on growing-season mean temperature and climatological yield responses does not allow capturing the year-to-year variations. Presumably capturing both effects simultaneously in a single statistical model would require different regressors than growing-season mean temperature. We have added language to clarify this point and to clarify that the emulators should not be used in the study of responses to short-term extremes within the growing season.

**Additional text pertaining to the choice of mean yields has been added to Section 2.2, Lines 118 - 125.**

4. Secondly, in Section 4.3, the authors demonstrate that potential shifts in the distributions of climate parameters do not impact the climatological yield emulations and the results still compare well with the GGCMs (Figure 9). This fact – this shift in distribution and potential changes in variability that result from it (which is where the readers' mind might go, as mine certainly did) – is I believe distinct from the discussion of year-to-year variability discussed above.

This is a separate but related point. Because our emulators are trained on climate simulations with uniform offsets and no change in the other moments of the distribution, we felt the need to show that the emulation could still faithfully capture the response of crop models when driven by a climate model projection, which includes some changes in variability. Because our emulator is not trained on any aspect

of year-over-year variability, it was important to ask whether changes in variability in climate models might be so large and impactful for crops that they dominated the effects of mean changes and made the GGCMI emulators not useful. By showing that the emulated yield change is equal to the change simulated under a climate projection with the same mean temperature shift, we demonstrate that any variability changes in climate projections are not large/impactful enough to invalidate the GGCMI emulators. We have added language to our discussion to clarify this point.

**Text modified in section 4.3, lines 396 - 415.**

5. I appreciate that the authors have provided detailed explanations of their approach, treatment, and findings wrt to considering these variability and distributional changes. Still, there's a lot of material here to keep track of, and I think it may be useful to reiterate each of the above points clearly in the Discussion (particularly if I've mistakenly represented it, as I think this may be an example of reader confusion!). For example, there is a sentence in the Discussion that minimizes the impact of future variability (par- ticularly at the aggregate level – around Line 445), particularly in the area aggregate, and I think this is in reference to the findings in Section 4.3 However, this doesn't mean that interannual variability, or extremes or nonlinear interactions, won't be impactful tofuture (or current) crop impacts.

We agree that a recap would be very helpful; thank you for the suggestion. We have expanded the discussion as suggested.

**Recap added to section 6, lines 496 - 527.**

6. Discussion: Lastly, I think the major point of this paper is to provide these emulator frameworks as an alternative to climate-crop assessments with the full process- based GGCMs. I therefore understand the authors' approach to evaluate the emulators against the GGCMs – this is quite reasonable.

It might be helpful, though, to take one step beyond this and compare to some observed historical yield changes. I would not expect this to be better than the GGCMs, and such evaluations have already been done for the GGCMs, so I would expect to see a similar response (and this is notwithstanding the applicability and veracity of comparison products). However, I don't think I've seen such an evaluation for GGCMI Phase 2 yet (I expect one is planned), and so pre-empting this with a comparison of the emulators may just be useful to have on hand. If this could be done and stuck into Supplementary, it would be a useful figure for the community moving forward, rather than having to show an intermediary figure of emulator-GGCM comparisons.

Unfortunately this type of validation is impossible with our current approach. On the decadal timescale, changes in management outweigh the effects of climate, and climate-driven mean yield changes in the historical record are impossible to disentangle from management changes. On the yearly timescale, as discussed above, the emulators are not appropriate for reproducing short-term variations, and so we also cannot use them to faithfully represent historical yearly yield anomalies (detrended from management changes).

The performance of the GGCMI Phase 2 crop models is addressed in the GMD companion paper (Franke et al. 2020a), using the standard evaluation approach based on the year-over-year time-series correlation

with FAO statistics (see Müller et al. 2017). However, this time-scale is not addressed by the emulators of the crop models, and so the emulators cannot be treated similarly.

[revised manuscript text omitted]
| LPJmL, von Bloh et al. (2018)                                                             | Х     | Х       | Х    | Х               | Х               | Х      | 756 / 648                  |
| pDSSAT , Elliott et al. (2014); Jones et al. (2003)                                | Х     | Х       | Х    | Х               | Х               | Х      | 756 / 648                  |
| PEPIC , Liu et al. (2016b, c)                                                      | Х     | Х       | Х    | Х               | Х               | Х      | 149 / 121                  |
| PROMET , Hank et al. (2015); Mauser et al. (2015); Zabel et al. (2019)             | Х     | Х       | Х    | Х               | Х               | _      | 261 / 232                  |

and the JULES model uses a bias-corrected version of ERA-Interim, WFDEI (WATCH-Forcing-Data-ERA-Interim, Weedon et al., 2014) as these groups have specific sub-daily input data requirements. Temperature perturbations are applied as additive mean shifts, water supply as fractional multipliers to precipitation (except in the irrigated  $W_{\infty}$  case), and CO2 and nitrogen

110 application levels are specified as fixed values. Models provide near-global output at 0.5 degree latitude and longitude resolution for each simulation year, including areas not currently cultivated. Crop models included here are not formally calibrated, given that there is no adequate calibration target for gridded global-scale crop model simulations. This may be a shortcoming if targeting absolute yield levels, but when focusing on relative yield changes, calibration can also have negative effects on model **Table 2.** GGCMI Phase  $H_2$  input levels for the parameter sweep. Values for temperature and water supply are perturbations from the historical climatology. For water supply, perturbations are fractional changes to historical precipitation, except in the irrigated ( $W_{\infty}$ ) simulations, which are all performed with the maximum beneficial levels of water. Bold font indicates the 'baseline' historical level. The full protocol samples across all parameter combinations for a total of 756 cases. Table repeated from Franke et al. (2020).

| Input variable       | Tested range                                                          | Unit          |
|----------------------|-----------------------------------------------------------------------|---------------|
| $[CO_2](C)$          | 360 , 510, 660, 810                                            | ppm           |
| Temperature (T)      | -1, 0 , 1, 2, 3, 4, 6                                          | °C            |
| Precipitation (W)    | -50, -30, -20, -10, 0 ,                                        | %             |
|                      | 10, 20, 30, (and $W_{\infty}$ )                                       |               |
| Applied nitrogen (N) | 10, 60, 200                                                    | kg ha $^{-1}$ |
| Adaptation (A)       | A0: none, A1: new cultivar to maintain original growing season length | -             |

skill (Müller et al., 2017). In analyses where we distinguish yields over currently cultivated land, we use the harvested area
masks of Portmann et al. (2010). (See Supplemental Figure S2 for maps of cultivated area.)

**2.2 Climatological vs. year-to-year responses**

We emulate the climatological mean response, because that is the response of interest. The central metric in assessments of climate change impacts . The on crop yields is the change in multi-annual means (e.g. Schlenker and Roberts, 2009; Challinor et al., 2014; Ro . Agricultural impacts assessments work with multi-annual yields, as their analysis frameworks require information on long-term

- 120 effects (e.g. Nelson et al., 2014b; Stevanović et al., 2016; Wiebe et al., 2015; Hasegawa et al., 2018; Snyder et al., 2019). Changes in extremes or year-to-year response can be significantly different from the forced climatological one, so we do not use information from year-to-year variability but instead emulate the are other metrics of potential interest, but are often not explicitly considered in integrated climate change impact assessments or land-use change projections. For this reason we emulate the climatological mean response, i.e. the change in aggregated mean yield in each 30-year simulation. Emulation
- 125 then becomes relatively straightforward, since changes in time-averaged yields are also-considerably smoother than those in year-to-year yield response.

In the GGCMI Phase H-2 simulation output dataset, year-to-year responses to weather are also often quantitatively distinct from responses to climatological shifts, with the discrepancy especially strong in wheat and rice. The difference in behavior is illustrated in Figure 1, which shows irrigated and rainfed maize and wheat in representative locations; open circles and black

130 lines show the climatological mean response, and solid circles and colored lines the responses for the 30 individual years in individual scenarios. When discrepancies are large, year-to-year responses are generally stronger than climatological ones, but exact responses differ by crop and region and even by model within GGCMI PhaseII. 2.